# A Graph Foundation Model with Cross-Modal Alignment and Modality-Aware Expert Fusion for Multi-Modal Graphs

Dongxiao He [1]  Ankang Yang [1]  Jitao Zhao [1]  Di Jin [1]

## Abstract

Graph Foundation Models (GFMs) aim to learn universal patterns through large-scale pretraining on diverse graphs and generalize to open-world scenarios. While GFMs have garnered significant attention, existing works primarily focus on single-modal graphs. However, many real-world graphs are multimodal, consisting of structures alongside diverse features derived from modalities such as text and images. To date, exploration into Multimodal Graph Foundation Models (MGFMs) remains limited. Incorporating multimodal data provides a more comprehensive view, allowing models to learn richer semantics, thereby advancing GFMs. We are therefore motivated to explore MGFMs, where the core challenge lies in synergistically encoding structures and multimodal features to achieve effective cross-modal alignment and fusion. To this end, we propose a graph foundation model with **C**ross-modal **A**lignment and **M**odality-aware **E**xpert fusion, CAME. Specifically, CAME first generates graph embeddings for each individual modality. We then introduce a multimodal multi-expert encoding mechanism, which includes a dimension-wise routing strategy to fuse multimodal information. Finally, we employ a cross-modal contrastive loss to train CAME, enabling the adaptive alignment and fusion across different modalities. Extensive experiments demonstrate the effectiveness of CAME across multiple tasks and diverse multimodal graph datasets.

## 1. Introduction

Graph data, consisting of nodes and edges, has great potential to represent the complex relationships ubiquitous in the real world (Wu et al., 2021; Zhou et al., 2020). Consequently, graphs are widely used in various applications, such as social networks analysis (Perozzi et al., 2014), e-commerce recommendation systems (Ying et al., 2018), and molecular predictions (Gilmer et al., 2017). Graph representation learning, which aims to model complex and irregular graph to generate effective representations for downstream tasks, stands as a central pillar of graph mining. Early methods primarily employed heuristic algorithms, such as matrix factorization (Koren et al., 2009) and random walks (Grover & Leskovec, 2016), to encode structure. Subsequently, Graph Neural Networks (GNNs) emerged as the dominant paradigm. By leveraging message-passing mechanisms or transformer architectures, GNNs can synergistically encode both features and structures to learn effective patterns (Zhuo et al., 2023; 2025).

Recently, inspired by the success of universal Large Language Models (LLMs), the research focus has gradually shifted from scenario-specific GNNs to universal Graph Foundation Models (GFMs) (Mao et al., 2024). GFM researchers strive to develop graph models that, through pretraining on massive graphs, acquire a comprehensive understanding of graph patterns and generalize robustly to unseen applications. Existing GFMs can be broadly categorized into LLM-based methods and graph-centric universal modeling methods. The former leverages LLMs to unify diverse semantic spaces, integrating their knowledge and reasoning capabilities with graph models (Tang et al., 2024; Wang et al., 2024; Li et al., 2024b). The latter, independent of LLMs, focuses on mining intrinsic patterns within graphs to universally model distinct structural and semantic patterns (Veličković et al., 2018; Thakoor et al., 2021; Hou et al., 2023). While effective, these works are primarily designed for graphs with unimodal features.

However, many real-world graphs are multimodal, where entities are associated with multi-source information such as text, images, and audio (Ektefaie et al., 2023; Peng et al., 2024). Few existing GFMs consider such common scenarios. Drawing inferences from the rapid evolution of

---

[1]School of Computer Science and Technology, Tianjin University, Tianjin, China. Correspondence to: Jitao Zhao <zjtao@tju.edu.cn>.

*Proceedings of the 43rd International Conference on Machine Learning*, Seoul, South Korea. PMLR 306, 2026. Copyright 2026 by the author(s).

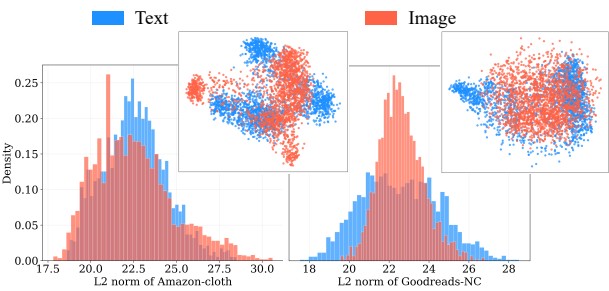

*Figure 1.* Modality distribution and visualization.

multimodal large models (Wu et al., 2023), Multimodal Graph Foundation Models (MGFMs) may become a pivotal frontier. This direction not only significantly expands the application scope of GFMs but also enhances their intrinsic capabilities. Because multimodal graphs provide a more comprehensive view of entities (Baltrusaitis et al., 2019), enabling the model to understand node semantics and relationships more comprehensively. This, in turn, fosters a more universal understanding of graph data, boosting performance in both unimodal and multimodal scenarios.

This motivated us to become one of the early efforts in MGFMs. We identify the core challenge as synergistically encoding structures with multimodal node features and achieving cross-modal alignment and fusion. To the best of our knowledge, UniGraph2 represents the only preliminary attempt (He et al., 2025b). However, UniGraph2 averages CLIP-encoded features from different modalities before structure propagation, which may collapse modality-specific information before graph modeling and hinder adaptive cross-modal alignment and fusion. Moreover, different scenarios possess different modal patterns. Imposing a shared encoding mechanism across modalities tends to overlook these differences, failing to identify differentiated contributions. We empirically validate this challenge by visualizing the text and image modalities of Goodreads-NC and Amazon-cloth, along with their L2-norm distributions, as shown in Figure 1. We observed that text-image feature co-occurrence patterns vary significantly across multimodal graphs. This discrepancy necessitates carefully designed strategies for multimodal graph alignment and fusion.

To solve these challenges, we propose a multimodal graph foundation model that performs **C**ross-modal **A**lignment with **M**odality-aware **E**xpert fusion (CAME). Specifically, we build a modality-specific graph encoder, allowing the model to adaptively encode the structures and features tailored to each modal distribution. To enhance fine-grained fusion, we introduce a dimension-wise gating mechanism that dynamically combines information from different modalities. We further propose a modality-aware Mixture-of-Experts module to learn the relationships among both inter

and intra modalities. Finally, we treat the fused representation as a distinct modality and design a cross-modal contrastive loss to align and fuse the embeddings of different modalities. Our contributions are summarized as follows:

- We explore the multimodal graph foundation models, expanding the scope of graph universality and fostering a more comprehensive understanding of graphs.

- We propose CAME, which leverages a dimension-wise gating mechanism and a modality-aware expert fusion module to effectively encode structures and multimodal features. We also propose a cross-modal contrastive loss to align and fuse multimodal embeddings.

- We conduct extensive experiments demonstrating that CAME not only exhibits robust generalizability on multimodal graphs but also outperforms existing GFMs on unimodal graphs.

**Conflict of Interest Disclosure.** The authors declare no financial or other substantive conflicts of interest related to this work.

## 2. Related Work

**Graph Foundation Models.** GFMs aim to learn generalizable graph representations via large-scale pretraining, enabling unified modeling and robust transfer across graphs, domains, and tasks (Mao et al., 2024; Zhuo et al., 2024a). Recent GFMs further study multi-dataset pretraining under domain shifts, focusing on coordinating cross-domain optimization to reduce negative transfer, preventing degradation caused by forced cross-dataset feature alignment, and improving fully-inductive generalization to unseen graphs with varying structures and attribute spaces (Zhao et al., 2024a;b;c). Meanwhile, with large language models, text-attributed graph GFMs have emerged: UniGraph unifies graphs across domains as TAGs and co-trains language models with graph encoders (He et al., 2025a), and OFA extends this unification with standardized task prompts and in-context learning (Liu et al., 2024). However, most existing GFMs are built upon unimodal node attributes and lack mechanisms for fine-grained multimodal fusion and cross-modal alignment under graph propagation, which limits their ability to learn robust multimodal graph representations.

**Multimodal Graph Learning.** Multimodal graph learning aims to jointly model graph structure and multimodal features to obtain more informative representations for downstream tasks (Ektefaie et al., 2023). A common design choice is how to couple multimodal fusion with graph propagation. Early-fusion methods fuse multimodal attributes before message passing and then propagate the fused features on the graph, which is widely used in multimodal KG

embedding and graph recommendation (Liu et al., 2023; Li et al., 2024a). Complementarily, propagation-aware methods couple message passing with cross-modal interaction by injecting multi-hop structural context into the cross-modal attention (Ning et al., 2025). However, most of these methods are developed for single-domain and single-task settings. Recently, several studies have begun to explore incorporating the foundation model paradigm into multimodal graph learning. For example, UniGraph2 introduces modality-specific masking and an expert-composition mechanism for multimodal graphs (He et al., 2025b). However, due to the distribution gap across modalities, it constructs the message-passing input via coarse aggregation of unaligned modality embeddings, which can limit fine-grained cross-modal interactions. Therefore, achieving fine-grained fusion with effective cross-modal alignment remains a key challenge for learning high-quality multimodal embeddings.

## 3. Preliminaries

**Multimodal Graph.** We consider a multimodal graph $\mathcal{G} = (\mathcal{V}, \mathcal{E})$ with $N = |\mathcal{V}|$ nodes, where $\mathcal{V}$ and $\mathcal{E}$ denote the node and edge sets, respectively (Zhu et al., 2024). Each node is associated with multiple modalities. For our work, we focus on two modalities: text $(t)$ and visual $(v)$. For a node $i \in \mathcal{V}$, its raw input in modality $m \in \{t, v\}$ is $\mathbf{s}_i^{(m)}$. The goal of multimodal graph representation learning is to learn node representations by jointly exploiting the graph structure and multimodal attributes for downstream tasks.

**Graph Neural Networks.** Graph neural networks commonly follow the message passing paradigm (Wu et al., 2020). Let $\mathbf{h}_i^{(l)}$ denote the representation of node $i$ at layer $l$, and let $\mathcal{N}(i)$ be its neighborhood. A general message passing layer can be expressed by aggregating the current-node representation together with transformed neighbor representations:

$$\mathbf{h}_i^{(l+1)} = \text{AGG}\Big(\mathbf{h}_i^{(l)}, \{\mathbf{W}^{(l)}\mathbf{h}_j^{(l)} \mid j \in \mathcal{N}(i)\}\Big), \quad (1)$$

where $\mathbf{W}^{(l)}$ is a learnable weight matrix shared across neighbors at layer $l$, and $\text{AGG}(\cdot)$ is an aggregation function that combines the center node with its neighborhood messages.

**Self-supervised Graph Learning.** Self-supervised graph learning aims to learn graph representations without labels by constructing supervision from the data itself. Among various self-supervised objectives, contrastive learning is a prevalent approach: it maximizes agreement between two augmented views of the same entity while separating different entities (Veličković et al., 2018; You et al., 2020; Zhuo et al., 2024b). Typically, two augmented views $\mathcal{G}^a$ and $\mathcal{G}^b$ are generated from the original graph. A shared encoder $f_\theta(\cdot)$ produces embeddings $\mathbf{z}_i^a$ and $\mathbf{z}_i^b$ for entity $i$.

Given a batch of $N$ entities, $(\mathbf{z}_i^a, \mathbf{z}_i^b)$ forms a positive pair,

while $\mathbf{z}_j^b$ for $j \neq i$ are treated as negatives. The InfoNCE objective is:

$$\mathcal{L}_{\text{InfoNCE}} = -\frac{1}{N}\sum_{i=1}^{N}\log\frac{\exp(\text{sim}(\mathbf{z}_i^a, \mathbf{z}_i^b))}{\sum_{j=1}^{N}\exp(\text{sim}(\mathbf{z}_i^a, \mathbf{z}_j^b))}, \quad (2)$$

This loss encourages view-invariant representations by pulling the same entity across views closer while separating different entities. For multimodal graphs, modalities naturally serve as alternative views, enabling us to impose cross-modal consistency.

## 4. Method

CAME learns fused multimodal graph representations under realistic modality gaps, where scale and distribution mismatches can make early fusion dilute information or be dominated by one modality. We first apply modality-specific message passing to obtain structure-aware unimodal views, then we introduce a dimension-wise gated fusion to enable fine-grained modal interaction and selectively integrate complementary dimensions. To handle varying fusion patterns across nodes and domains, a modality-aware expert fusion sparsely composes experts within each view and adaptively combines unimodal and fused outputs. Finally, a tri-view contrastive objective aligns text, vision, and fusion representations to reduce unimodal dominance.

### 4.1. Single-Modality Encoding

We consider a multimodal graph $\mathcal{G} = (\mathcal{V}, \mathcal{E})$ with $N = |\mathcal{V}|$ nodes. Each node $i \in \mathcal{V}$ is associated with two modalities of raw inputs: a text description and an image, denoted by $\mathbf{s}_i^{(t)}$ and $\mathbf{s}_i^{(v)}$, respectively. In realistic multimodal graphs, text and vision features often follow different statistics, so early fusion may blur modality-specific cues. We thus encode each modality separately into a comparable embedding space, providing clean unimodal views for later fusion and alignment. Therefore, we first employ a pretrained CLIP model to obtain modality embeddings (Radford et al., 2021):

$$\mathbf{x}_i^{(t)} = f_{\text{CLIP}}^{(t)}(\mathbf{s}_i^{(t)}), \qquad \mathbf{x}_i^{(v)} = f_{\text{CLIP}}^{(v)}(\mathbf{s}_i^{(v)}), \quad (3)$$

where $f_{\text{CLIP}}^{(t)}(\cdot)$ and $f_{\text{CLIP}}^{(v)}(\cdot)$ denote the CLIP text encoder and image encoder, respectively. Since the CLIP text and image encoders output embeddings with the same dimensionality, we have $\mathbf{x}_i^{(t)}, \mathbf{x}_i^{(v)} \in \mathbb{R}^d$. This shared embedding space enables subsequent gating, expert modules, and contrastive objectives to be applied directly across modalities.

Feature distributions can differ substantially across modalities, making it difficult for a single neighborhood aggregator to model modality-specific neighborhood patterns. Moreover, scale mismatches and dispersion mismatches between modality features can cause early fusion followed by shared

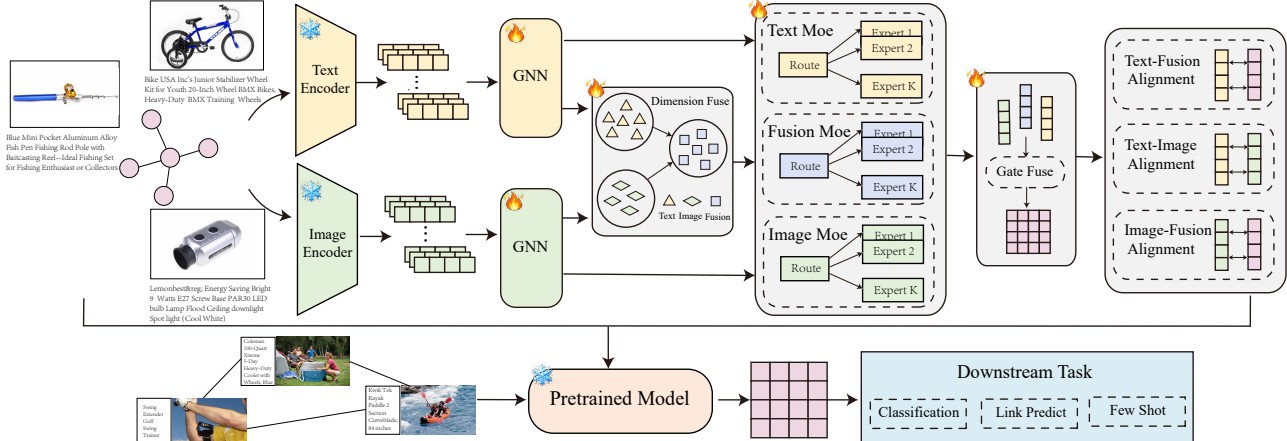

*Figure 2.* Overview of CAME.

propagation to be dominated by one modality, leading to information dilution and unstable transfer. To avoid such entanglement, we adopt modality-specific graph encoders: one GNN for text features and another GNN for visual features (Velickovic et al., 2018). Each encoder performs message passing on the same topology $\mathcal{G}$ but with its own modality features.

$$\mathbf{z}_i^{(t)} = f_{\text{GNN}}^{(t)}\Big(\mathcal{G},\ \{\mathbf{x}_j^{(t)}\}_{j \in \mathcal{V}}\Big), \tag{4}$$

$$\mathbf{z}_i^{(v)} = f_{\text{GNN}}^{(v)}\Big(\mathcal{G},\ \{\mathbf{x}_j^{(v)}\}_{j \in \mathcal{V}}\Big), \tag{5}$$

Here $f_{\text{GNN}}^{(t)}$ and $f_{\text{GNN}}^{(v)}$ can be any standard GNN backbone. $\mathbf{z}_i^{(t)}$ and $\mathbf{z}_i^{(v)}$ denote the representations obtained by aggregating neighborhood information for the center node $i$ under the text and visual modalities, respectively. By separating the two propagation, the model can preserve modality-specific structural signals before cross-modal fusion, providing cleaner unimodal views for our subsequent dimension-wise gating and modality-aware expert fusion, and enabling more stable cross-modal alignment during pretraining.

To make this modality-specific propagation practical on large-scale graphs, we train on compact PPR-sampled subgraphs rather than performing full-graph message passing (Jeh & Widom, 2003; Klicpera et al., 2019). This sampling tends to include informative multi-hop neighbors while keeping computation bounded, and we run all modality-specific encoders on each sampled subgraph.

### 4.2. Dimension-wise Gated Fusion

CAME's goal is to learn a fused node representation that integrates both the graph structure and complementary multimodal semantics. A single scalar weight is too coarse to handle dimension-level noise, and naive concatenation can

exacerbate modality scale mismatch. Instead of using a single scalar weight or naive concatenation, we introduce a dimension-wise gate that adaptively controls the contribution of each modality at each dimension. Specifically, we compute a gating vector $\mathbf{g}_i$ from the unimodal embeddings:

$$\mathbf{g}_i = \sigma\Big(\mathbf{W}_g[\mathbf{z}_i^{(t)}\|\mathbf{z}_i^{(v)}]\Big), \tag{6}$$

where $[\cdot\|\cdot]$ is concatenation, $\odot$ is element-wise and $\sigma(\cdot)$ is the element-wise sigmoid. We then form a gated fusion embedding:

$$\mathbf{z}_i^{(g)} = \mathbf{g}_i \odot \mathbf{z}_i^{(t)} + (1 - \mathbf{g}_i) \odot \mathbf{z}_i^{(v)}, \tag{7}$$

where $\odot$ is element-wise and $\mathbf{g}_i$ is the dimension-wise gate. The gate can emphasize the modality that is more informative for a given dimension and suppress unreliable coordinates from the other modality, reducing the impact of modality scale or dispersion mismatch. Consequently, the fused embedding retains complementary semantics while being more robust across domains.

**Comparison to scalar gating: an MSE-optimal view.** The discussion above suggests that fusion should adapt at the coordinate level rather than via a single global weight. We now provide a simple MSE-optimal perspective showing that, even under a basic noise model, the optimal linear fusion is generally dimension-dependent. Eq. (7) can be seen as a diagonal instance of a linear fusion family. We view $\mathbf{z}_i^{(t)}$ and $\mathbf{z}_i^{(v)}$ as two unbiased but noisy estimates of an underlying signal $\mathbf{h}_i$, and consider

$$\hat{\mathbf{h}}_i(\mathbf{G}) = \mathbf{G}\mathbf{z}_i^{(t)} + (\mathbf{I} - \mathbf{G})\mathbf{z}_i^{(v)}. \tag{8}$$

Our dimension-wise gate corresponds to restricting $\mathbf{G}$ to be diagonal, $\mathbf{G} = \text{diag}(\mathbf{g}_i)$.

**Proposition 4.1.** *Let $\Sigma_t, \Sigma_v$ be the noise covariances and $\Sigma_{tv}$ the cross-covariance, and define $\mathbf{M} := \Sigma_t + \Sigma_v - \Sigma_{tv} - \Sigma_{tv}^\top \succ \mathbf{0}$. Then the MSE minimizer of Eq. (8) is*

$$\mathbf{G}^\star = (\Sigma_v - \Sigma_{tv}^\top) \mathbf{M}^{-1}. \tag{9}$$

*Remark* 4.2. Proposition 4.1 indicates that the MSE-optimal fusion is generally *dimension-sensitive* (and in general not even restricted to a scalar or diagonal form). In particular, a scalar gate corresponds to $\mathbf{G} = \alpha\mathbf{I}$, which is a much more constrained subfamily. This motivates learning a diagonal, dimension-wise gate in Eq. (7) as a lightweight yet expressive approximation.

### 4.3. Modality-Aware MoE

While the gated embedding $\mathbf{z}_i^{(g)}$ provides a direct and fine-grained mixture of text and vision, it still implements a single parametric fusion rule shared across nodes. In realistic multimodal graphs, the informative modality and the desired fusion pattern can vary substantially across nodes and graphs. Therefore, we adopt a modality-aware expert fusion with modality-specific expert groups, where different experts can specialize to different fusion behaviors and be activated sparsely per node. Specifically, our MoE adopts a two-level routing design: an intra-group router enables modality-specific expert composition, and a second-stage router aggregates across groups to adaptively balance unimodal signals and fused semantics under distribution shifts.

We maintain three separate expert groups: text experts ($t$), image experts ($v$), and fusion experts ($g$). Each group contains $K$ experts:

$$\mathcal{X}^{(s)} = \{\phi_1^{(s)}, \ldots, \phi_K^{(s)}\}, \qquad s \in \{t, v, g\}, \tag{10}$$

Text and image experts perform representation transformations within their respective modalities to preserve modality-specific characteristics; fusion experts further explore interactions between text and vision, complementing associations that are difficult to capture with a single modality alone. We use $\mathbf{z}_i^{(t)}$, $\mathbf{z}_i^{(v)}$, and $\mathbf{z}_i^{(g)}$ as the inputs to the three MoE groups, respectively.

**Intra-Modal Interactions.** Within each expert group $s \in \{t, v, g\}$, we perform sparse routing by selecting the experts that best match node $i$. This design allows different experts to specialize in different patterns within the same modality, rather than forcing a single shared transformation for all nodes. Concretely, a group-specific gating network $g^{(s)}(\cdot)$ produces routing logits

$$\mathbf{r}_i^{(s)} = g^{(s)}\left(\mathbf{z}_i^{(s)}\right) \in \mathbb{R}^K, \tag{11}$$

where $K$ is the number of experts in group $s$. We then select the top-$k$ experts with indices $\mathcal{S}_i^{(s)} = \text{TopK}(\mathbf{r}_i^{(s)}, k)$

and compute normalized weights over the selected set via a softmax restricted to $\mathcal{S}_i^{(s)}$:

$$\alpha_{i,j}^{(s)} = \frac{\exp\left(r_{i,j}^{(s)}\right)}{\sum\limits_{q \in \mathcal{S}_i^{(s)}} \exp\left(r_{i,q}^{(s)}\right)}, \qquad j \in \mathcal{S}_i^{(s)}. \tag{12}$$

Importantly, we adopt top-$k$ expert evaluation: only experts in $\mathcal{S}_i^{(s)}$ are executed for node $i$. The group output is computed as

$$\mathbf{e}_i^{(s)} = \sum_{j \in \mathcal{S}_i^{(s)}} \alpha_{i,j}^{(s)} \, \phi_j^{(s)}(\mathbf{z}_i^{(s)}). \tag{13}$$

Top-$k$ routing enforces sparse expert activation, which encourages specialization. At the same time, activating a small set of experts retains the capacity for adaptive composition, allowing the model to integrate complementary semantics.

**Inter-Modal Interactions.** In multimodal graphs, different nodes rely on different modalities to varying degrees: some nodes are well explained by text semantics, while others require visual cues. Under cross-modal and cross-domain shifts, these preferences can further change due to scale mismatch and distribution drift. Therefore, beyond intra-group expert composition, we introduce a modality-level router to decide *which expert group(s)* should dominate for each node. Specifically, we compute modality-level weights that indicate the relevance of each group for node $i$:

$$\boldsymbol{\pi}_i = [\pi_i^{(t)}, \pi_i^{(v)}, \pi_i^{(g)}] = \text{Softmax}(g(\mathbf{z}_i)), \tag{14}$$

here $\mathbf{z}_i = [\mathbf{z}_i^{(t)}\|\mathbf{z}_i^{(v)}\|\mathbf{z}_i^{(g)}]$. $g(\cdot)$ is typically implemented as MLP. By assigning weights to the three expert groups through modality-level routing, we improve the flexibility of fusion and enhance robustness under cross-domain settings. Finally, we aggregate the three group outputs using the modality-level weights:

$$\mathbf{z}_i^{(f)} = \pi_i^{(t)}\mathbf{e}_i^{(t)} + \pi_i^{(v)}\mathbf{e}_i^{(v)} + \pi_i^{(g)}\mathbf{e}_i^{(g)}. \tag{15}$$

Overall, the modality-aware expert fusion operates at multiple granularities: intra-group routing learns modal-specific expert compositions, while the modality-level router adaptively balances unimodal and fused groups. This design yields flexible cross-modal integration and improves stability under distribution shifts.

### 4.4. Cross-Modal Contrastive Loss

While our gating and MoE modules enable fine-grained fusion, multimodal graphs still exhibit significant modality gaps: text and vision features can have mismatched scales and shifted distributions, and the fused representation may drift toward the dominant modality if it is trained without

alignment. Aligning only a single pair (e.g., text–vision) can reduce the gap between the two unimodal spaces, but it does not guarantee that the fused space is compatible with both modalities. To encourage transferability and modality consistency, we jointly align the three embedding spaces $\mathbf{z}_i^{(t)}$, $\mathbf{z}_i^{(v)}$, and $\mathbf{z}_i^{(f)}$ through an InfoNCE-style contrastive objective. We define similarity with temperature $\tau$ as

$$\mathrm{sim}(\mathbf{p}, \mathbf{q}) = \frac{\mathbf{p}^\top \mathbf{q}}{\tau}, \tag{16}$$

For any pair of distinct views $\{a, b\} \subset \{t, v, f\}$, we treat $(\mathbf{z}_i^{(a)}, \mathbf{z}_i^{(b)})$ as the positive pair for node $i$, and all $\{(\mathbf{z}_i^{(a)}, \mathbf{z}_j^{(b)})\}_{j \neq i}$ as negatives:

$$\mathcal{L}_{a,b} = -\frac{1}{N} \sum_{i=1}^{N} \log \frac{\exp(\mathrm{sim}(\mathbf{z}_i^{(a)}, \mathbf{z}_i^{(b)}))}{\sum_{j=1}^{N} \exp(\mathrm{sim}(\mathbf{z}_i^{(a)}, \mathbf{z}_j^{(b)}))}, \tag{17}$$

We apply the contrastive loss to all three pairs among $\{t, v, f\}$: $(t, v)$, $(v, f)$, and $(t, f)$.

$$\mathcal{L} = w_{tv}\mathcal{L}_{t,v} + w_{vf}\mathcal{L}_{v,f} + w_{tf}\mathcal{L}_{t,f}, \tag{18}$$

where $w_{tv}, w_{vf}, w_{tf}$ are hyperparameters. The first term enforces direct alignment between unimodal structural embeddings, while the latter two terms encourage the fused space to remain compatible with each unimodal space. Intuitively, $\mathcal{L}_{t,v}$ promotes direct correspondence between unimodal structural embeddings, mitigating the modality gap after message passing. The other two terms, $\mathcal{L}_{t,f}$ and $\mathcal{L}_{v,f}$, encourage the fused space to remain compatible with each unimodal space. This tri-view alignment stabilizes fusion, discourages unimodal dominance, and yields a more transferable unified representation for downstream tasks.

**A rank-based justification.** Let $\mathbf{u}_i^{(a)}$ be the $\ell_2$-normalized embedding from view $a \in \{t, v, f\}$, and define $\ell_{ij}^{ab} := \langle \mathbf{u}_i^{(a)}, \mathbf{u}_j^{(b)} \rangle / \tau$. For an anchor $i$, we define the (tie-conservative) retrieval rank of its positive pair as

$$\mathrm{rank}_{ab}(i) := 1 + \#\{j \neq i : \ell_{ij}^{ab} \geq \ell_{ii}^{ab}\}. \tag{19}$$

**Proposition 4.3.** *Fix any $K \in \{1, \ldots, N-1\}$. If the per-sample InfoNCE loss satisfies $\ell_{ab,i}^{\mathrm{NCE}} < \log(K + 1)$, then $\mathrm{rank}_{ab}(i) \leq K$.*

*Remark* 4.4. Proposition 4.3 implies that minimizing InfoNCE encourages the positive pair to be retrieved within the top-$K$ matches (with respect to the sampled negatives). Applying it to both $(t, f)$ and $(v, f)$ suggests that the fused embedding $\mathbf{z}_i^{(f)}$ remains simultaneously retrievable from text and vision, which helps prevent unimodal dominance.

## 5. Experiments

### 5.1. Baseline

We categorize the compared methods into two groups: self-supervised contrastive learning methods and graph foundation models. The self-supervised contrastive learning methods include DGI (Veličković et al., 2018), GRACE (Zhu et al., 2020), BGRL (Thakoor et al., 2021) and GraphMAE2 (Hou et al., 2023), which are typically pre-trained on a single graph and learn node representations through contrastive or reconstruction-based objectives. Graph foundation models include GCOPE (Zhao et al., 2024a), FUG (Zhao et al., 2024b), SAMGPT (Yu et al., 2025), and UniGraph2 (He et al., 2025b). They adopt multi-graph pre-training: a shared encoder is jointly trained on a corpus of graphs spanning multiple datasets, and then directly transferred to downstream tasks without extra pre-training.

### 5.2. Experimental Setting

For text-only datasets, we use a placeholder visual input by reusing the CLIP text embedding for the missing image modality, so that all methods receive inputs in the same format. For multimodal datasets, except for UniGraph2, since the other models do not support multimodal inputs, we uniformly use CLIP to encode the text and images, and then concatenate the resulting text and image features as the input, so as to ensure a fair comparison. All baseline GFMs are re-trained from scratch under the same multi-graph pre-training corpus and protocol as CAME. In Self-Supervised Pre-training Evaluation, models are jointly pre-trained on Amazon-cloth, Goodreads-LP, and Products, and then evaluated on the corresponding datasets for node classification, edge classification, and link prediction. For few-shot evaluation, we use models pretrained on Amazon-cloth, Goodreads-NC, and Products. (Hu et al., 2020; Zhu et al., 2024). Detailed settings and hyperparameters are provided in Appendix C.

### 5.3. Self-Supervised Pre-training Evaluation

Table 1 reports that CAME achieves the best performance across all datasets and evaluation settings. Compared with baseline methods that only use CLIP to encode multi-modal features, as well as classical graph self-supervised approaches such as BGRL, DGI, GraphMAE2, and GRACE (Thakoor et al., 2021; Veličković et al., 2018; Hou et al., 2023; Zhu et al., 2020), CAME consistently demonstrates superior performance on all tasks. These results indicate that, in multimodal graph scenarios, simply relying on uni-modal encoders or directly applying standard graph self-supervised pre-training paradigms is insufficient to fully capture the complementary relationships among heterogeneous information sources within multimodal neighborhoods.

*Table 1.* Results for self-supervised representation learning. We report accuracy (%) for node/edge classification and MRR (%) for link prediction. We consider in-distribution, in-domain, and out-of-domain settings based on whether the test dataset/domain is seen during pre-training. The best method in each column is bolded, and the runner-up is underlined.

| | In-distribution | | | In-domain Generalization | | | Out-of-domain Generalization | | | |
|---|---|---|---|---|---|---|---|---|---|---|
| | Products | Goodreads-LP | Amazon-cloth | Amazon-sports | Goodreads-NC | Ele-fashion | Arxiv | Wiki-CS | FB15K237 | WN18RR |
| **Use CLIP to encode raw multimodal data as input features.** | | | | | | | | | | |
| BGRL | 72.57±0.05 | 6.49±0.02 | 17.35±0.03 | 23.66±0.05 | 69.28±0.10 | 74.92±0.03 | 65.07±0.04 | 71.35±0.02 | 90.12±0.15 | 75.02±0.14 |
| DGI | 71.91±0.06 | 7.23±0.07 | 18.39±0.04 | 24.07±0.11 | 70.39±0.08 | 83.72±0.02 | 61.07±0.02 | 69.79±0.16 | 91.43±0.17 | 73.26±0.11 |
| GraphMAE2 | 70.46±0.15 | 7.77±0.12 | 16.25±0.11 | 27.44±0.14 | 72.49±0.05 | 80.23±0.04 | 64.15±0.18 | 74.82±0.19 | 89.96±0.14 | 73.97±0.14 |
| GRACE | 73.44±0.08 | 4.77±0.02 | 18.15±0.05 | 25.59±0.05 | 67.47±0.12 | 79.70±0.02 | 58.11±0.04 | 67.28±0.02 | 90.12±0.15 | 73.02±0.19 |
| GCOPE | 76.87±0.14 | 8.91±0.11 | 17.49±0.12 | 24.24±0.13 | 75.43±0.15 | 77.61±0.14 | 63.92±0.09 | 75.94±0.18 | 91.03±0.16 | 75.61±0.13 |
| SAMGPT | 78.53±0.11 | 9.13±0.06 | 19.14±0.06 | 28.45±0.08 | 79.02±0.09 | 83.82±0.07 | 69.79±0.17 | 74.63±0.18 | 91.76±0.10 | 74.96±0.15 |
| FUG | 74.34±0.17 | 7.94±0.10 | 20.49±0.11 | 25.33±0.12 | 74.62±0.14 | 81.29±0.01 | 68.11±0.05 | 73.31±0.12 | 91.49±0.12 | 74.35±0.10 |
| **Use raw multimodal data as input features.** | | | | | | | | | | |
| CLIP-text | 64.83±0.09 | 3.71±0.09 | 13.77±0.09 | 25.13±0.06 | 69.45±0.07 | 83.53±0.02 | 62.75±0.03 | 64.32±0.19 | 89.72±0.21 | 74.86±0.13 |
| CLIP-image | – | 2.77±0.07 | 13.09±0.07 | 16.33±0.02 | 64.29±0.12 | 80.26±0.05 | – | – | – | – |
| UniGraph2 | 80.82±0.08 | 8.76±0.06 | 24.25±0.12 | 29.52±0.14 | 79.78±0.07 | 84.83±0.11 | 70.54±0.04 | 77.07±0.06 | 92.30±0.03 | 75.20±0.09 |
| CAME | **81.84±0.17** | **10.42±0.08** | **28.88±0.12** | **37.68±0.19** | **83.21±0.11** | **85.13±0.03** | **71.66±0.05** | **78.57±0.13** | **94.59±0.07** | **81.97±0.11** |

*Table 2.* Results for few-shot transfer. We report accuracy (%) for node classification. We consider in-distribution, in-domain, and out-of-domain settings based on whether the test dataset/domain is seen during pre-training. The best method in each column is bolded, and the runner-up is underlined.

| | In-distribution | | | Out-of-domain Generalization | | | | | |
|---|---|---|---|---|---|---|---|---|---|
| | Goodreads-NC-5-way | | | Arxiv-5-way | | | Wiki-CS-5-way | | |
| | 1-shot | 3-shot | 5-shot | 1-shot | 3-shot | 5-shot | 1-shot | 3-shot | 5-shot |
| **Use CLIP to encode raw multimodal data as input features.** | | | | | | | | | |
| BGRL | 28.73±9.16 | 39.32±11.45 | 44.19±11.88 | 38.93±11.02 | 54.79±12.26 | 59.57±12.28 | 33.98±10.40 | 44.75±11.22 | 49.86±11.98 |
| DGI | 26.34±8.65 | 33.67±10.28 | 38.00±10.63 | 40.49±11.26 | 55.23±11.91 | 61.93±11.44 | 31.55±9.23 | 45.04±10.60 | 51.65±10.96 |
| GraphMAE2 | 25.21±7.94 | 32.11±9.68 | 37.22±10.64 | 37.48±10.67 | 56.67±12.14 | 64.31±11.74 | 31.92±12.72 | 43.39±11.71 | 50.78±11.13 |
| GRACE | 38.95±11.22 | 48.71±11.73 | 52.48±11.66 | 41.02±11.14 | 54.69±12.56 | 61.17±12.17 | 36.14±12.43 | 48.38±11.50 | 53.79±10.39 |
| GCOPE | 37.60±11.35 | 48.88±10.17 | 54.34±9.91 | 47.50±12.89 | 56.10±12.35 | 59.91±12.46 | 39.04±12.96 | 53.26±11.04 | 59.53±10.77 |
| SAMGPT | 29.95±10.09 | 34.96±10.36 | 37.65±10.63 | 48.37±12.43 | 59.57±11.34 | 64.26±11.26 | 39.52±11.73 | 49.38±11.28 | 54.02±10.78 |
| FUG | 42.76±12.61 | 50.49±12.86 | 53.09±12.68 | 47.20±13.06 | 56.77±12.88 | 60.27±12.51 | 32.32±10.15 | 46.83±11.16 | 49.40±11.48 |
| **Use raw multimodal data as input features.** | | | | | | | | | |
| CLIP-text | 34.94±10.75 | 45.73±11.22 | 49.26±12.19 | 42.30±11.25 | 58.72±12.12 | 65.03±11.27 | 32.81±9.70 | 46.95±10.87 | 54.31±10.63 |
| CLIP-image | 27.33±8.65 | 35.55±10.29 | 39.95±11.32 | – | – | – | – | – | – |
| UniGraph2 | 39.66±11.33 | 53.06±11.77 | 58.00±11.65 | 43.11±11.90 | 60.74±11.96 | 66.96±11.13 | 35.56±10.04 | 50.95±11.04 | 58.12±10.54 |
| CAME | **48.03±12.54** | **60.26±12.30** | **60.47±11.01** | **49.38±13.18** | **63.78±12.17** | **67.10±11.67** | **47.65±12.86** | **55.30±11.92** | **61.89±12.95** |

*Table 3.* Ablation on CAME key components.

| Setting | Ele-fashion | Goodreads-NC | Amazon-cloth | Amazon-sports |
|---|---|---|---|---|
| CAME | **85.13±0.03** | **83.21±0.11** | **28.88±0.12** | **37.68±0.19** |
| w/o $L_{t,v}$ | 84.67±0.04 | 82.46±0.09 | 20.64±0.15 | 34.30±0.12 |
| w/o $L_{t,f}$ | 85.04±0.03 | 83.13±0.07 | 27.27±0.10 | 37.24±0.16 |
| w/o $L_{v,f}$ | 82.44±0.07 | 82.58±0.11 | 24.62±0.14 | 36.26±0.10 |
| only $L_{t,v}$ | 83.95±0.05 | 82.20±0.09 | 21.66±0.17 | 32.99±0.13 |
| only $L_{t,f}$ | 80.74±0.11 | 80.70±0.12 | 19.19±0.11 | 34.82±0.18 |
| only $L_{v,f}$ | 84.35±0.04 | 81.48±0.08 | 22.03±0.10 | 35.12±0.14 |
| w/o Moe | 84.39±0.03 | 82.55±0.06 | 26.37±0.12 | 36.36±0.10 |
| w/o dim-gate | 84.75±0.06 | 82.26±0.07 | 27.90±0.13 | 37.41±0.09 |
| w/o m-GNN | 84.52±0.04 | 81.92±0.06 | 21.19±0.13 | 26.83±0.11 |

Across both in-domain and out-of-domain settings, CAME clearly outperforms existing graph foundation models, including GCOPE, SAMGPT, FUG and UniGraph2 (Zhao et al., 2024a; Yu et al., 2025; Zhao et al., 2024b; He et al., 2025b). This indicates that, under a unified training protocol,

CAME is able to learn more transferable representations and effectively support knowledge transfer across tasks and datasets within the same domain. As shown in Fig 3, we further conduct t-SNE visualizations on the Ele-fashion and Goodreads-NC datasets, projecting high-dimensional features into a 2D space (Maaten & Hinton, 2008). The results show that the representations learned by CAME exhibit clearer inter-class boundaries and more compact intra-class clusters in the 2D space, leading to a more stable clustering structure. These observations further validate the importance of modeling modality-aware structural propagation and fine-grained multimodal fusion during message passing.

### 5.4. Few-shot Evaluation

The experimental setup for the few-shot setting is described in Appendix C. During inference, classification is performed by matching query samples to class prototypes, which are

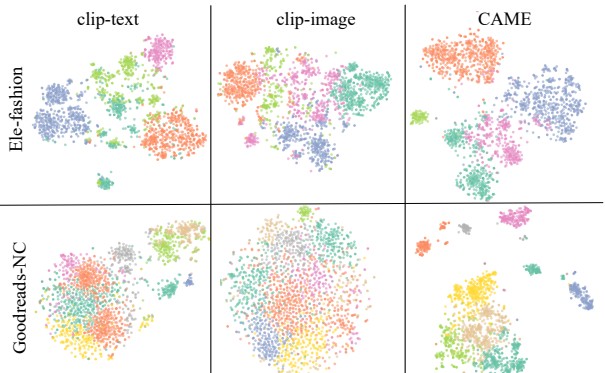

*Figure 3.* t-SNE visualizations.

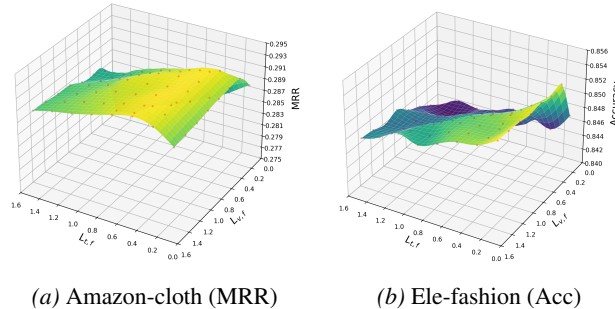

*(a)* Amazon-cloth (MRR)     *(b)* Ele-fashion (Acc)

*Figure 4.* Evaluation metric varies with $(L_{t,f}, L_{v,f})$.

computed as the mean representations of the support sets. Table 2 reports the few-shot performance under the 1-shot, 3-shot, and 5-shot settings. Overall, CAME achieves the best results on all datasets and all shot numbers, showing strong label efficiency and robust transfer. Notably, CAME brings large improvements on Goodreads-NC, where multimodal signals are available. This suggests that even with extremely limited labeled data, CAME can still produce a stable fused embedding space.

On the two text-only benchmarks (Arxiv and Wiki-CS), CAME still delivers substantial gains over CLIP features, indicating its strong generality: although CAME is designed for multimodal graphs, it does not rely on the presence of multiple modalities and can generalize well to unimodal graphs. This indicates that the improvements are not merely from stronger raw encoders, but from learning structure-aware and transferable node representations. CAME leverages neighborhood aggregation to inject graph context, and adaptive expert composition to model diverse local structural patterns, producing an embedding space that is more discriminative for prototype-based few-shot classification—even when only a single modality is available.

### 5.5. Model Analysis

Table 3 reports the ablation study of loss and key components. Removing any alignment loss consistently degrades performance, indicating that the three consistency constraints are complementary. In particular, removing $\mathcal{L}_{t,v}$ often leads to the most significant drop, while removing $\mathcal{L}_{t,f}$ has the smallest impact. This suggests that the textual representations are already of high quality, and during joint training the final fused representation can further exploit complementary semantics from the visual modality, yielding high quality multimodal embeddings. Moreover, keeping only a single alignment loss performs noticeably worse, confirming that a single alignment signal is insufficient to ensure cross-modal semantic consistency and a well-structured fused space. We consider three ablation settings of key

components: w/o m-GNN, which performs early fusion by element-wise mean pooling of the two modalities, using the averaged representation as the input to a single shared GNN for message passing and aggregation; w/o dim-gate, which replaces the dimension-wise gated fusion with simple average pooling over text and image representations; and w/o MoE, which substitutes the modality-aware expert fusion module with a parameter-matched MLP fusion. All three variants lead to performance drops of varying magnitudes across datasets, suggesting that keeping modality-specific message passing and fine-grained gated fusion is beneficial, and that the modality-aware expert fusion's two-stage routing further improves fusion adaptivity and the resulting multimodal embedding quality.

### 5.6. Loss-Weight Sensitivity

As illustrated in Fig 4, we conduct experiments on the Ele-fashion and Amazon-cloth by fixing $L_{t,v}$ and systematically varying the weights of $L_{t,f}$ and $L_{v,f}$, and then visualize the model performance as a 3D surface. The results show that performance does not change monotonically with the loss weights; instead, a relatively stable high-performance region emerges at moderate weight values. Practically, this indicates that CAME is not overly sensitive to precise weighting, and a moderate range of $(L_{t,f}, L_{v,f})$ reliably balances fused-space consistency with unimodal discriminability. When the alignment constraint between a single modality and the fused representation is either too strong or too weak, the performance drops to varying degrees. These results suggest that the text–fusion and vision–fusion constraints play complementary roles during training: proper alignment helps mitigate cross-modal distribution gaps and stabilizes the semantic structure of the fused representation, whereas overly strong constraints may suppress the intra-modal discriminative capacity. The results validate the necessity of adopting the tri-view consistency constraints in CAME.

### 5.7. Conclusions

We propose CAME, a graph foundation model with cross-modal alignment and modality-aware expert fusion for

multi-modal graphs. It employs modality-specific GNN encoders to model structural patterns for different modalities. CAME then performs a two-stage fusion with a dimension-wise gate and a modality-aware expert fusion module, enabling fine-grained fusion and flexible intra-modal expert integration with gated inter-modal fusion. A cross-modal contrastive loss further aligns the fused representation. Extensive results on ten benchmarks across in-distribution, in-domain, and out-of-domain settings demonstrate consistent improvements over strong baselines, validating CAME's effectiveness and transferability.

## Impact Statement

"This paper presents work whose goal is to advance the field of Machine Learning. There are many potential societal consequences of our work, none which we feel must be specifically highlighted here."

## Acknowledgments

This work was supported by the National Natural Science Foundation of China (No. 62422210, No. 62276187, No. 92370111, and No. 62272340).

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

# A. Proofs

## A.1. Proof of Proposition 4.1 (MSE-optimal linear fusion)

**Proposition A.1.** *Let $\Sigma_t, \Sigma_v$ be the noise covariances and $\Sigma_{tv}$ the cross-covariance, and define $\mathbf{M} := \Sigma_t + \Sigma_v - \Sigma_{tv} - \Sigma_{tv}^\top \succ 0$. Then the MSE minimizer of Eq. (8) is*

$$\mathbf{G}^\star = (\Sigma_v - \Sigma_{tv}^\top)\, \mathbf{M}^{-1}. \tag{9}$$

*Proof.* Recall the linear fusion family

$$\hat{\mathbf{h}}(\mathbf{G}) = \mathbf{G}\mathbf{z}^{(t)} + (\mathbf{I} - \mathbf{G})\mathbf{z}^{(v)}, \qquad \mathbf{z}^{(m)} = \mathbf{h} + \boldsymbol{\varepsilon}^{(m)}.$$

Then

$$\hat{\mathbf{h}}(\mathbf{G}) - \mathbf{h} = \mathbf{G}\boldsymbol{\varepsilon}^{(t)} + (\mathbf{I} - \mathbf{G})\boldsymbol{\varepsilon}^{(v)}.$$

Let $\Sigma_t = \mathbb{E}[\boldsymbol{\varepsilon}^{(t)}\boldsymbol{\varepsilon}^{(t)\top}]$, $\Sigma_v = \mathbb{E}[\boldsymbol{\varepsilon}^{(v)}\boldsymbol{\varepsilon}^{(v)\top}]$, and $\Sigma_{tv} = \mathbb{E}[\boldsymbol{\varepsilon}^{(t)}\boldsymbol{\varepsilon}^{(v)\top}]$. The MSE objective is

$$\mathcal{J}(\mathbf{G}) = \mathbb{E}\left[\left\|\mathbf{G}\boldsymbol{\varepsilon}^{(t)} + (\mathbf{I} - \mathbf{G})\boldsymbol{\varepsilon}^{(v)}\right\|_2^2\right].$$

Expanding and taking trace form yields

$$\mathcal{J}(\mathbf{G}) = \mathrm{Tr}\Big(\mathbf{G}\Sigma_t\mathbf{G}^\top + (\mathbf{I} - \mathbf{G})\Sigma_v(\mathbf{I} - \mathbf{G})^\top + \mathbf{G}\Sigma_{tv}(\mathbf{I} - \mathbf{G})^\top + (\mathbf{I} - \mathbf{G})\Sigma_{tv}^\top\mathbf{G}^\top\Big).$$

Differentiate w.r.t. $\mathbf{G}$ and set the gradient to zero:

$$\mathbf{G}(\Sigma_t + \Sigma_v - \Sigma_{tv} - \Sigma_{tv}^\top) = \Sigma_v - \Sigma_{tv}^\top.$$

Define $\mathbf{M} := \Sigma_t + \Sigma_v - \Sigma_{tv} - \Sigma_{tv}^\top$. Under the assumption $\mathbf{M} \succ 0$, $\mathbf{M}$ is invertible and thus

$$\mathbf{G}^\star = (\Sigma_v - \Sigma_{tv}^\top)\, \mathbf{M}^{-1},$$

which matches Eq. (9). $\qquad\square$

**Diagonal (dimension-wise) intuition.** If we restrict $\mathbf{G} = \mathrm{diag}(\mathbf{g})$ and additionally assume coordinate-wise independent noises and cross-modality independence, the MSE decouples over dimensions. Each coordinate $j$ reduces to a 1D convex quadratic, whose minimizer is

$$g_j^\star = \frac{\sigma_{v,j}^2}{\sigma_{t,j}^2 + \sigma_{v,j}^2}.$$

This shows the optimal fusion generally prefers *different weights across dimensions*, aligning with our learned dimension-wise gate in Eq. (7).

## A.2. Proof of Proposition 4.3 (Rank-based implication of InfoNCE)

**Proposition A.2.** *Fix any $K \in \{1, \ldots, N - 1\}$. If the per-sample InfoNCE loss satisfies $\ell_{ab,i}^{\mathrm{NCE}} < \log(K + 1)$, then $\mathrm{rank}_{ab}(i) \leq K$.*

*Proof.* Fix $i$ and denote $\ell_{ii}^{ab}$ by $\ell^+$. If $\mathrm{rank}_{ab}(i) \geq K + 1$, then there exist at least $K$ indices $j \neq i$ such that $\ell_{ij}^{ab} \geq \ell^+$. Hence the InfoNCE denominator satisfies

$$\sum_{j=1}^N \exp(\ell_{ij}^{ab}) \geq \exp(\ell^+) + \sum_{\substack{j \neq i: \\ \ell_{ij}^{ab} \geq \ell^+}} \exp(\ell_{ij}^{ab}) \geq \exp(\ell^+) + K\exp(\ell^+) = (K + 1)\exp(\ell^+).$$

Therefore,

$$\ell_{ab,i}^{\mathrm{NCE}} = -\log \frac{\exp(\ell^+)}{\sum_{j=1}^N \exp(\ell_{ij}^{ab})} \geq -\log \frac{\exp(\ell^+)}{(K + 1)\exp(\ell^+)} = \log(K + 1),$$

contradicting $\ell_{ab,i}^{\mathrm{NCE}} < \log(K + 1)$. Thus $\mathrm{rank}_{ab}(i) \leq K$. $\qquad\square$

## B. Algorithm

---

**Algorithm 1** CAME Pre-Training

---

1: **Input:** Graph $\mathcal{G} = (\mathcal{V}, \mathcal{E})$, $N = |\mathcal{V}|$;
2:     raw texts $\{\mathbf{s}_n\}_{n=1}^N$, raw images $\{\mathbf{I}_n\}_{n=1}^N$.
3: **Output:** Parameters $\Theta$ (and optionally fused embeddings $\mathbf{Z}^{(f)}$).
4: **(1) Modality encoders:** raw text/image $\rightarrow$ node features
5:     $\mathbf{X}^{(t)} \leftarrow \text{TextEnc}(\{\mathbf{s}_n\}_{n=1}^N) \in \mathbb{R}^{N \times d_t}$
6:     $\mathbf{X}^{(v)} \leftarrow \text{ImageEnc}(\{\mathbf{I}_n\}_{n=1}^N) \in \mathbb{R}^{N \times d_v}$
7: **while** not converged **do**
8:     **for** each mini-batch of nodes $\mathcal{B} \subseteq \mathcal{V}$ (or full graph) **do**
9:       **(2) Graph encoders:** propagate modality features over graph
10:       $\mathbf{Z}_{\mathcal{B}}^{(t)} \leftarrow \text{GAT}_t(\mathcal{G}_{\mathcal{B}}, \mathbf{X}^{(t)})\big|_{\mathcal{B}}$
11:       $\mathbf{Z}_{\mathcal{B}}^{(v)} \leftarrow \text{GAT}_v(\mathcal{G}_{\mathcal{B}}, \mathbf{X}^{(v)})\big|_{\mathcal{B}}$
12:       **(3) Dimension-wise gate :** produce gated fusion view
13:       $\boldsymbol{g}_{\mathcal{B}} \leftarrow \sigma\Big(\text{MLP}_g([\mathbf{Z}_{\mathcal{B}}^{(t)}; \mathbf{Z}_{\mathcal{B}}^{(v)}])\Big)$
14:       $\mathbf{Z}_{\mathcal{B}}^{(g)} \leftarrow \boldsymbol{g}_{\mathcal{B}} \odot \mathbf{Z}_{\mathcal{B}}^{(t)} + (1 - \boldsymbol{g}_{\mathcal{B}}) \odot \mathbf{Z}_{\mathcal{B}}^{(v)}$
15:       **(4) Modality-Aware expert fusion:** two-level routing
16:       $\mathbf{H}_{\mathcal{B}} \leftarrow \Big[\mathbf{Z}_{\mathcal{B}}^{(t)} \| \mathbf{Z}_{\mathcal{B}}^{(v)} \| \mathbf{Z}_{\mathcal{B}}^{(g)}\Big]$
17:       $\boldsymbol{\pi}_{\mathcal{B}} \leftarrow \text{Softmax}(\text{Router}_{\text{mod}}(\mathbf{H}_{\mathcal{B}}))$
18:       *(where $\boldsymbol{\pi}$ is over {text, image, fusion})*
19:       $\mathbf{U}_{\mathcal{B}}^{(t)} \leftarrow \text{TopK-MoE}(\mathbf{Z}_{\mathcal{B}}^{(t)}, \{\text{Expert}_j^{(t)}\}_{j=1}^K, \text{Router}_{\text{exp}}^{(t)}, k)$
20:       $\mathbf{U}_{\mathcal{B}}^{(v)} \leftarrow \text{TopK-MoE}(\mathbf{Z}_{\mathcal{B}}^{(v)}, \{\text{Expert}_j^{(v)}\}_{j=1}^K, \text{Router}_{\text{exp}}^{(v)}, k)$
21:       $\mathbf{U}_{\mathcal{B}}^{(g)} \leftarrow \text{TopK-MoE}(\mathbf{Z}_{\mathcal{B}}^{(g)}, \{\text{Expert}_j^{(g)}\}_{j=1}^K, \text{Router}_{\text{exp}}^{(g)}, k)$
22:       $\mathbf{Z}_{\mathcal{B}}^{(f)} \leftarrow \pi_{\mathcal{B}}^{(t)} \odot \mathbf{U}_{\mathcal{B}}^{(t)} + \pi_{\mathcal{B}}^{(v)} \odot \mathbf{U}_{\mathcal{B}}^{(v)} + \pi_{\mathcal{B}}^{(g)} \odot \mathbf{U}_{\mathcal{B}}^{(g)}$
23:       **(5) Cross-Modal contrastive losses**
24:       $\mathcal{L}_{tv} \leftarrow \text{InfoNCE}(\mathbf{Z}_{\mathcal{B}}^{(t)}, \mathbf{Z}_{\mathcal{B}}^{(v)}; \tau)$
25:       $\mathcal{L}_{vf} \leftarrow \text{InfoNCE}(\mathbf{Z}_{\mathcal{B}}^{(v)}, \mathbf{Z}_{\mathcal{B}}^{(f)}; \tau)$
26:       $\mathcal{L}_{tf} \leftarrow \text{InfoNCE}(\mathbf{Z}_{\mathcal{B}}^{(t)}, \mathbf{Z}_{\mathcal{B}}^{(f)}; \tau)$
27:       $\mathcal{L} \leftarrow w_{tv}\mathcal{L}_{tv} + w_{vf}\mathcal{L}_{vf} + w_{tf}\mathcal{L}_{tf}$
28:       Update $\Theta$ by minimizing $\mathcal{L}$ (e.g., Adam).
29:     **end for**
30: **end while**
31: **return** $\Theta$ (and optionally $\mathbf{Z}^{(f)}$)

---

Algorithm 1 summarizes the overall pre-training procedure of CAME. Given a multimodal graph $\mathcal{G} = (\mathcal{V}, \mathcal{E})$, each node is associated with raw textual and visual content. In Step (1), frozen CLIP encoders are applied to transform raw texts and images into initial modality embeddings $\mathbf{x}_i^{(t)}$ and $\mathbf{x}_i^{(v)}$, as defined in Eq. (3). In Step (2), modality-specific GNN encoders propagate unimodal features over the same graph structure. Specifically, two independent GNNs produce structure-enhanced embeddings $\mathbf{z}_i^{(t)}$ and $\mathbf{z}_i^{(v)}$ according to Eqs. (4)–(5), preserving modality-specific neighborhood information before fusion. In Step (3), a dimension-wise gated fusion view is constructed. A gating network computes $\mathbf{g}_i$ from the concatenation of unimodal embeddings via Eq. (6), and the gated representation $\mathbf{z}_i^{(g)}$ is obtained via Eq. (7), enabling fine-grained text–vision interaction at the embedding dimension level. In Step (4), CAME performs modality-aware expert fusion. It maintains three expert groups (text/image/fusion) as in Eq. (10), and within each group $s \in t, v, g$, sparse top-$k$ routing is performed by computing routing logits $\mathbf{r}_i^{(s)}$ (Eq. (11)), normalized weights $\alpha i, j^{(s)}$ (Eq. (12)), and the group output $\mathbf{e}_i^{(s)}$ (Eq. (13)). A modality-level router then assigns weights $\boldsymbol{\pi}_i$ over the three expert groups (Eq. (14)), and the final fused representation $\mathbf{z}_i^{(f)}$ is obtained by inter-group aggregation in Eq. (15). In Step (5), cross-modal contrastive learning is applied to align the unimodal and fused embedding spaces. Similarity is defined in Eq. (16), the InfoNCE loss for any view pair $(a, b)$ is given

in Eq. (17), and CAME applies it to all three pairs $(t, v)$, $(t, f)$, and $(v, f)$, combining them into the overall objective $\mathcal{L}$ in Eq. (18) All model parameters $\Theta$ are optimized end-to-end by minimizing the overall contrastive loss. After convergence, the learned fused node embeddings can be directly reused for downstream tasks.

## C. Implementation Details

*Table 4.* Pre-training hyper-parameters.

| $L_{t,v}$ | $L_{t,f}$ | $L_{v,f}$ | hidden_size | lr | dropout | num_epochs | num_gnn_layers | ppr_topk | num_experts | selected_experts |
|---|---|---|---|---|---|---|---|---|---|---|
| 0.5 | 0.1 | 0.5 | 1024 | 1e-4 | 0.3 | 8 | 2 | 128 | 6 | 2 |

**Running environment.** All experiments are conducted on a Linux server with 125 GiB RAM and 8 NVIDIA GeForce RTX 5090 GPUs (32 GB memory each). We use Python 3.9.23, PyTorch 2.8.0 (CUDA 12.8), and DGL 1.1.3.

**Baselines.** We use the official released implementations for all open-source baselines and follow their recommended default settings. For a fair comparison, we keep the same dataset splits and evaluation protocols across methods, and report mean performance over multiple random seeds. For CLIP-based feature baselines, we extract node features using frozen pretrained encoders and directly evaluate them with the same downstream protocols, without adding additional graph message passing or fusion components.

**Pre-training details.** We initialize text and image features using frozen CLIP encoders (Radford et al., 2021). CAME is pretrained for 8 epochs with learning rate $1 \times 10^{-4}$ and dropout 0.3. All modality-specific GNN encoders use 2 message-passing layers (num_gnn_layers=2) and produce 1024-dimensional hidden representations (hidden_size=1024). To scale message passing to large graphs, we train on PPR-sampled subgraphs with top-128 PPR neighbors per seed node. Our modality-aware expert fusion contains 6 experts per view (num_experts=6), and uses sparse top-$k$ routing with $k = 2$ activated experts per node (selected_experts=2). For the tri-view contrastive objective, we set the pairwise loss weights to $(w_{tv}, w_{tf}, w_{vf}) = (0.5, 0.1, 0.5)$, which balances direct text–image alignment with fused-space consistency.

**Data splits and evaluation protocol.** We follow the official splits for all datasets to ensure comparability with prior work. For ogbn-Arxiv and ogbn-Products, we use the standard OGB splits and report results over 10 seeds (Hu et al., 2020). For Wiki-CS, we use the 20 official training splits (each containing $5\%$ labeled nodes per class) and report averages over splits with 20 random initializations (Liu et al., 2024). For knowledge graph datasets (FB15K-237 and WN18RR), we adopt the same splits as OFA and average over 10 seeds (Liu et al., 2024). For Amazon-sports, Amazon-cloth, Goodreads-LP, Goodreads-NC, and Ele-fashion, we use the official splits and run 10 seeds (Zhu et al., 2024). Across all settings, hyperparameters are selected using the validation set, and we apply early stopping based on validation performance.

**Knowledge graph evaluation.** For knowledge graph datasets (FB15K-237 and WN18RR), we evaluate pretrained node representations via a relation classification protocol rather than ranking-based link prediction (Liu et al., 2024). Each triplet $(h, r, t)$ is treated as an edge classification instance. We freeze the pretrained encoder, form an edge representation by concatenating head and tail embeddings $[\mathbf{z}_h \| \mathbf{z}_t]$, and train a linear classifier to predict relation $r$ with cross-entropy loss. Model selection is performed by validation-based early stopping, and we report test accuracy.

**Linear probing.** To evaluate representation quality under a standard transfer protocol, we perform linear probing with a frozen encoder. We first extract and fix node embeddings for all nodes, then train a linear classifier on the training split. We use learning rate 0.05 and train for up to 2000 epochs with validation-based early stopping. The classifier parameters $\mathbf{W} \in \mathbb{R}^{d \times |\mathcal{Y}|}$ are optimized by minimizing cross-entropy loss on the labeled training nodes, and we report the best validation-selected test performance.

**Few-shot transfer.** We follow the episodic evaluation protocol of UniGraph2 (He et al., 2025b) under the same $N$-way $K$-shot setup. For each episode, we sample $N$ classes, select $K$ labeled nodes per class as the support set, and evaluate on the remaining labeled nodes (queries) from the same $N$ classes. During inference, we compute a class prototype as the mean embedding of its support nodes, and classify each query node by matching it to the nearest class prototype in the embedding

space. We freeze the pretrained encoder throughout few-shot evaluation, repeat each configuration with 10 random seeds, and report the averaged accuracy.

## D. Datasets

*Table 5.* Statistics of all 10 multimodal graph datasets.

| Dataset | Domain | Task | #Nodes | #Edges | Raw Features |
|---|---|---|---|---|---|
| ogbn-Arxiv | Citation | Node | 169,343 | 1,166,243 | Paper Titles and Abstracts |
| ogbn-Products | Product | Node | 2,449,029 | 61,859,140 | Product Descriptions |
| Wiki-CS | Wikipedia | Node | 11,701 | 216,123 | Wikipedia Entry Names and Contents |
| Ele-fashion | Product | Node | 97,766 | 199,602 | Fashion Titles and Fashion Images |
| Goodreads-NC | Book | Node | 685,294 | 7,235,084 | Book Descriptions and Book Images |
| FB15K237 | Knowledge | Edge | 14,541 | 310,116 | Entity Names and Descriptions |
| WN18RR | Knowledge | Edge | 40,943 | 93,003 | Entity Names and Descriptions |
| Amazon-sports | Product | Edge | 50,250 | 356,202 | Product Titles and Product Images |
| Amazon-cloth | Product | Edge | 125,839 | 951,271 | Product Titles and Product Images |
| Goodreads-LP | Book | Edge | 636,502 | 3,437,017 | Book Descriptions and Book Images |

**Text-only graphs.** We include three text-attributed graphs where each node is associated with raw textual descriptions and no visual modality is used. OGBN-ARXIV is a directed citation network of computer science arXiv papers, where nodes correspond to papers and directed edges denote citations. The downstream task is node classification over 40 subject areas. Following OGB, we use the concatenation of paper titles and abstracts as node text features (Hu et al., 2020). OGBN-PRODUCTS is an undirected Amazon co-purchase graph, where nodes are products and edges connect products that are frequently bought together. The downstream task is node classification with 47 product categories. We adopt the official OGB text features derived from product titles/descriptions (Hu et al., 2020). WIKI-CS is a Wikipedia hyperlink graph, where nodes are pages and edges represent reference links between pages. The downstream task is node classification over page categories. We use page names and page contents as node text features as collected in OFA (Liu et al., 2024). We further consider knowledge graphs where nodes are entities connected by typed relations (edges). We evaluate on edge classification, and construct node text features using entity names and textual descriptions provided by OFA (Liu et al., 2024). FB15K-237 is a Freebase-derived KG consisting of relation triples, with the standard 237-relation split that removes inverse-relation redundancy. WN18RR is a WordNet-derived KG containing 11 relation types and 40,943 entities, designed to avoid test leakage caused by reversible relations. Since our framework focuses on self-supervised node representation learning and relation prediction is used only as the downstream objective, we only use *node* text features and do not incorporate edge text features, which could otherwise introduce trivial shortcuts or potential information leakage (Liu et al., 2024).

**Multimodal graphs.** We include five multimodal graphs where each node is paired with both text and an image, following the multimodal graph benchmark construction in (Zhu et al., 2024). In these datasets, the text modality is the item title and/or description, while the visual modality is the associated product image or book cover. All graphs are built from user interaction signals (e.g., co-purchase or co-interest), and the downstream tasks cover both node classification and link prediction. Amazon-sports and Amazon-cloth are category-specific Amazon product graphs, where nodes are items in the sports or clothing category and edges encode co-purchase relations. Each node includes the product title (and description when available) as text and the main product image as vision input (Zhu et al., 2024). Goodreads-LP and Goodreads-NC are Goodreads book graphs, where nodes are books and edges capture user-driven co-interest signals (e.g., books frequently shelved or interacted with by the same users). Goodreads-LP is evaluated on link prediction and Goodreads-NC is evaluated on node classification. Each book is associated with textual metadata (title/description) and a cover image; items without images are removed to ensure consistent multimodal inputs across nodes (Zhu et al., 2024). Ele-fashion is a fashion product graph, where nodes are items and edges reflect purchase affinity or co-occurrence patterns. Each node contains the product title/description and a corresponding item image, and the downstream task is node classification (Zhu et al., 2024).

*Table 6.* Comparison of GPU memory, average joint pre-training time per epoch on ogbn-Products, Goodreads-LP, and Amazon-Cloth, downstream inference time on Goodreads-NC, and the number of parameters.

| Method | Memory (GB) | Pre-train (min) | Inference (min) | Params (M) |
|---|---|---|---|---|
| CAME | 14.67 | 2.13 | 6.9 | 53.6 |
| FUG | 18.45 | 13.09 | 11.2 | 10.5 |
| UniGraph2 | 28.37 | 11.58 | 10.3 | 32.5 |

# E. Complexity Analysis

For the original graph $G = (V, E)$, we train on a PPR-sampled subgraph $G_s = (V_s, E_s)$ per step, where all forward computations and the objective are computed on $V_s$ and $E_s$. Let $d$ be the (shared) embedding/hidden dimension, $h$ the number of attention heads (with $d_h = d/h$), $K$ the number of experts per expert group, $k$ the number of selected experts, and $B$ the loss batching size. In our implementation, the text, image, and fused representations all have dimension $d$. Each GAT layer has complexity $\mathcal{O}(|V_s| d_{\text{in}} d + |E_s| d)$ (linear projection plus edge-wise attention/aggregation). Since $d_{\text{in}} = d$ and we use two GAT layers per modality for two modalities, the overall encoding cost is $\mathcal{O}\big(4|V_s|d^2 + 4|E_s|d\big)$. If the fusion gate is enabled, an additional MLP on concatenated embeddings introduces $\mathcal{O}(|V_s|d^2)$ computation.

Our modality-aware expert fusion contains three expert groups (text/image/fusion) with $K$ experts each. In our implementation, top-$k$ experts are selected *before* expert evaluation, and only the selected experts are executed. Specifically, for each node we first compute routing scores over $K$ experts and perform top-$k$ selection, which costs $\mathcal{O}(3|V_s|(d_f K + K))$ if the router is a linear classifier (projection plus top-$k$), and then evaluate only $k$ experts per group. Each expert is a two-layer MLP $d_f \to d \to d$, so the expert computation costs $\mathcal{O}(3k|V_s|(d_f d + d^2))$. Finally, combining the selected experts' outputs incurs $\mathcal{O}(3|V_s|kd)$, which is typically dominated by the expert MLPs when $d$ is moderate.

For the contrastive objective without graph-structure loss, InfoNCE computes all-pairs similarities within the sampled subgraph, costing $\mathcal{O}(|V_s|^2 d)$ time per embedding pair; using anchor mini-batches of size $B$ (i.e., `loss_batch_size=` $B$) reduces the peak logits memory from $\mathcal{O}(|V_s|^2)$ to $\mathcal{O}(B|V_s|)$ while keeping the overall time complexity $\mathcal{O}(|V_s|^2 d)$. Overall, the forward pass on $G_s$ is dominated by $\mathcal{O}\big(|V_s|(d_t d + d_c d + 2d^2) + 2|E_s|d + 3k|V_s|(d_f d + d^2)\big)$ (plus router/top-$k$ overhead $\mathcal{O}(3|V_s|d_f K)$), and training additionally incurs $\mathcal{O}(|V_s|^2 d)$ from contrastive similarity computation (with peak logits memory $\mathcal{O}(B|V_s|)$).

We further report empirical efficiency and scalability results in Table 6, including GPU memory usage, average joint pre-training time per epoch, downstream inference time, and the number of parameters. For a fair comparison, we use the same hidden dimension for all compared methods. The pre-training time is measured on the joint pre-training datasets, including ogbn-Products, Goodreads-LP, and Amazon-Cloth, while the inference time is measured on Goodreads-NC.

As shown in Table 6, CAME achieves the shortest pre-training time among all compared methods while maintaining competitive GPU memory usage and inference efficiency. Although CAME contains more parameters than several baselines, most additional parameters come from modality-specific expert groups. Due to sparse top-$k$ routing, only a small subset of experts is activated for each input, so the increased model capacity does not lead to a proportional increase in computation. This is consistent with the complexity analysis above and allows CAME to improve representation capacity while keeping the actual training and inference cost manageable.

Several designs contribute to this efficiency. First, CAME uses offline precomputed PPR subgraphs, avoiding repeated neighborhood expansion during training. Second, for the large-scale ogbn-Products graph, we adopt subset pre-training to reduce per-epoch cost while preserving representative structural and semantic information. Third, the modality-aware expert fusion module uses sparse top-$k$ routing, activating only the most relevant experts instead of evaluating all experts. These designs jointly enable efficient pre-training.

To further verify scalability, we conduct node classification on the large-scale ogbn-Products graph, which contains 2,449,029 nodes and 61,859,140 edges. The successful application of CAME to this graph demonstrates that the proposed framework can scale to large multimodal graph scenarios.

*Table 7.* Notations used in our method.

| Symbol | Description |
|---|---|
| $\mathcal{G} = (\mathcal{V}, \mathcal{E})$ | Multimodal graph with nodes $\mathcal{V}$ and edges $\mathcal{E}$. |
| $\mathbf{s}_i^{(t)}, \mathbf{s}_i^{(v)}$ | Raw text and raw image of node $i$. |
| $\mathbf{x}_i^{(t)}, \mathbf{x}_i^{(v)} \in \mathbb{R}^d$ | CLIP features for text / vision modalities. |
| $\mathbf{z}_i^{(t)}, \mathbf{z}_i^{(v)} \in \mathbb{R}^d$ | Structure-aware embeddings from text / vision branches. |
| $\mathbf{g}_i$ | Dimension-wise gate for combining two modalities. |
| $\mathbf{z}_i^{(g)} \in \mathbb{R}^d$ | Gated fusion embedding (fusion view). |
| $\mathcal{X}^{(s)}$ | Expert group for view $s \in \{t, v, g\}$ |
| $\phi_i^{(s)}(\cdot)$ | The $i$-th expert in group $s$ ranked by routing score. |
| $K, k$ | $K$ experts per group and top-$k$ executed experts. |
| $\mathbf{z}_i^{(f)} \in \mathbb{R}^d$ | Final fused embedding after MoE fusion. |
| $\mathcal{L}_{t,v}, \mathcal{L}_{t,f}, \mathcal{L}_{v,f}$ | Pairwise InfoNCE losses among text, vision, and fused views. |
| $w_{t,v}, w_{t,f}, w_{v,f}$ | Weights for the three contrastive losses. |
| $\mathbf{h}_i^{(m)} \in \mathbb{R}^d$ | Embedding of node $i$ under modality/view $m$. |
| $\mathbf{r}_i \in \mathbb{R}^K$ | Routing score for node $i$ over experts. |
| $\mathcal{S}_i$ | Selected expert set for node $i$ (top-$k$ experts). |

## F. Analysis of Modality-Aware Expert Fusion

We further analyze the modality-aware expert fusion module in CAME. This module assigns one expert group to each modality-specific view, including text, image, and fused views, and uses a modality-level gating strategy to adaptively balance their contributions. This design allows CAME to preserve modality-specific patterns while modeling cross-modal interactions. We first examine the modality-level gate weights during training. The learned weights remain stable: the text weight changes from 0.48 to 0.42, the image weight from 0.02 to 0.09, and the fusion weight from 0.50 to 0.49. This indicates that CAME mainly relies on text and fused representations, while gradually increasing the contribution of image features. Although the image weight is smaller, its increase suggests that visual information provides complementary signals during training. To further understand expert specialization, we analyze expert activation frequency during the early stage of pre-training, as shown in Table 8. Different datasets exhibit distinct activation patterns across text, image, and fusion groups. For example, in the text group, all three datasets frequently activate E3 and E4, but with different relative frequencies. In the image group, ogbn-Products activates E2 and E5 more frequently, while Amazon-Cloth activates E1 and E6 more frequently. These observations indicate that expert groups do not collapse into a single dominant expert; instead, different experts specialize in different modality and dataset patterns.

We further report the selected experts during inference in Table 9. The selected experts vary across datasets and modalities, showing that CAME can adaptively route inputs to suitable experts according to their domain and modality characteristics. Moreover, inference-time selections are not always identical to early-stage activation patterns during pre-training, suggesting that CAME learns sufficiently specialized experts and further adapts expert selection to downstream data. Overall, these results demonstrate that modality-aware expert fusion enables domain-aware and modality-aware specialization.

## G. Robustness to Missing and Corrupted Modalities

We further analyze the robustness of CAME under missing and corrupted modality settings. In real-world multimodal graphs, modality information can be incomplete or noisy; therefore, a robust multimodal graph foundation model should handle missing modalities and maintain stable performance under modality corruption. CAME can be naturally applied to modality-missing datasets. In the main experiments, we evaluate CAME on text-only datasets such as Arxiv and WikiCS. Since the image modality is unavailable, we replace the missing image modality with the available text modality. CAME still consistently improves downstream performance, indicating that it does not strictly rely on all modalities being present and can adapt its routing and fusion mechanism to the available inputs. We further evaluate robustness under modality corruption by injecting Gaussian noise with different ratios into Ele-Fashion. Table 10 reports results under three settings: text-only noise, image-only noise, and both-modalities noise. As the noise ratio increases, CAME degrades gradually rather than collapsing, showing robustness to noisy multimodal inputs.

*Table 8.* Expert activation frequency during pre-training across datasets and modalities.

| Type | Dataset | E1 | E2 | E3 | E4 | E5 | E6 |
|------|---------|-----|-----|-----|-----|-----|-----|
| Text | Products | 0.5 | 1.2 | 35.0 | 31.3 | 6.8 | 25.2 |
| Text | Goodreads-NC | 0.6 | 0.2 | 33.8 | 36.1 | 6.3 | 23.1 |
| Text | Cloth | 0.3 | 0.1 | 33.1 | 27.0 | 6.3 | 33.1 |
| Image | Products | 11.1 | 43.4 | 1.2 | 1.1 | 31.0 | 12.2 |
| Image | Goodreads-NC | 18.6 | 26.4 | 2.0 | 0.7 | 14.7 | 37.8 |
| Image | Cloth | 26.3 | 15.3 | 1.0 | 0.2 | 13.6 | 43.6 |
| Fusion | Products | 38.7 | 10.7 | 2.3 | 0.7 | 15.9 | 31.7 |
| Fusion | Goodreads-NC | 37.4 | 8.6 | 6.6 | 2.3 | 13.3 | 31.7 |
| Fusion | Cloth | 36.3 | 11.0 | 3.9 | 3.1 | 14.1 | 31.6 |

*Table 9.* Selected experts at inference across datasets and modalities.

| Type | Dataset | Selected Experts |
|------|---------|------------------|
| Text | Amazon-Cloth | E3, E4 |
| Text | Arxiv | E4, E6 |
| Text | WN18RR | E1, E2 |
| Image | Amazon-Cloth | E1, E6 |
| Image | Arxiv | E3, E4 |
| Image | WN18RR | E2, E6 |
| Fusion | Amazon-Cloth | E1, E6 |
| Fusion | Arxiv | E2, E3 |
| Fusion | WN18RR | E2, E6 |

*Table 10.* Modality corruption results on Ele-Fashion. Gaussian noise with different ratios is injected into different modalities.

| Radio | 0.1 | 0.2 | 0.3 | 0.4 | 0.5 |
|-------|-----|-----|-----|-----|-----|
| Text-only | 84.43±0.05 | 84.00±0.06 | 83.61±0.23 | 83.17±0.10 | 82.73±0.11 |
| Image-only | 84.75±0.10 | 84.59±0.12 | 84.48±0.13 | 84.34±0.10 | 84.08±0.10 |
| Both | 84.30±0.06 | 83.48±0.08 | 82.88±0.11 | 82.34±0.08 | 81.38±0.22 |

The results also reflect the relative importance of different modalities on Ele-Fashion. Text-only noise causes a larger drop than image-only noise; for example, at the noise ratio of 0.5, text-only noise reduces performance to 82.73, while image-only noise maintains 84.08. This suggests that text is the more informative modality on this dataset, whereas image mainly provides complementary information. When both modalities are corrupted, the degradation is larger, reaching 81.38 at the noise ratio of 0.5. Overall, these results demonstrate that CAME can effectively exploit reliable modality signals and remain stable under modality corruption.

## H. Hyperparameter Analysis

The sensitivity results show that CAME maintains stable performance across a relatively wide range of loss weights, suggesting that our two-stage fusion (dimension-wise gating and modality-aware expert fusion) is not overly sensitive to precise hyperparameter choices and that tri-modal consistency primarily acts as a stabilizing regularizer. In particular, moderate $L_{t,v}$ typically yields consistent gains, which aligns with its role in mitigating the text–vision modality gap and improving cross-modal semantic correspondence, thereby making subsequent expert routing more reliable. This observation is also consistent with the scale and distribution shifts between text and image features revealed by the norm statistics and PCA projections on Ele-fashion and Amazon-Cloth. In contrast, overly large $L_{t,f}$ or $L_{v,f}$ often degrades performance, indicating that forcing the fused embedding to match a unimodal space too aggressively can suppress complementary modality signals and structural cues, leading to information dilution or single-modality dominance. Overall, balanced loss weights enable effective alignment while preserving modality-specific information, resulting in more robust generalization.

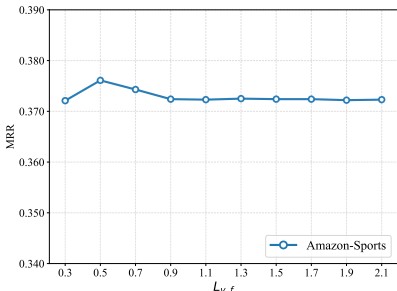

*Figure 5.* Amazon-sports: $L_{vf}$ hyperparameter analysis.

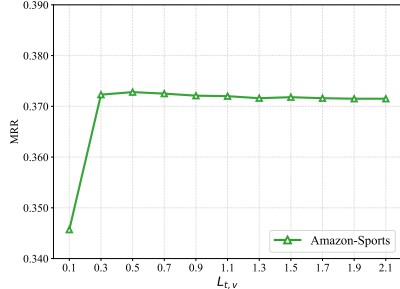

*Figure 6.* Amazon-sports: $L_{tv}$ hyperparameter analysis.

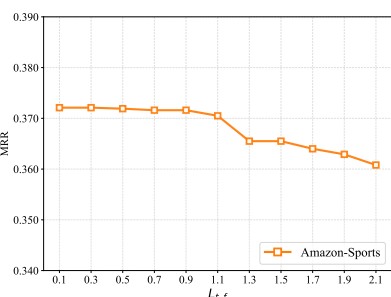

*Figure 7.* Amazon-sports: $L_{tf}$ hyperparameter analysis.

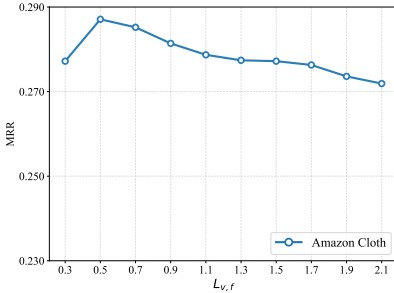

*Figure 8.* Amazon-cloth: $L_{vf}$ hyperparameter analysis.

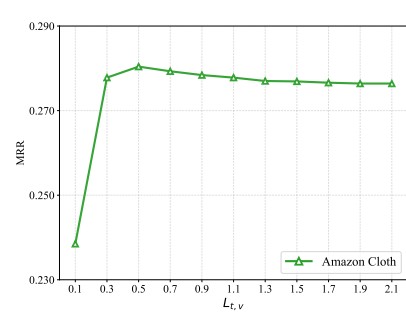

*Figure 9.* Amazon-cloth: $L_{tv}$ hyperparameter analysis.

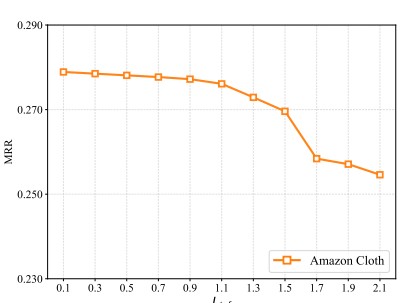

*Figure 10.* Amazon-cloth: $L_{tf}$ hyperparameter analysis.

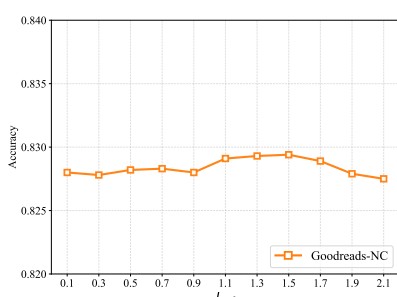

*Figure 11.* Goodreads-NC: $L_{tf}$ hyperparameter analysis.

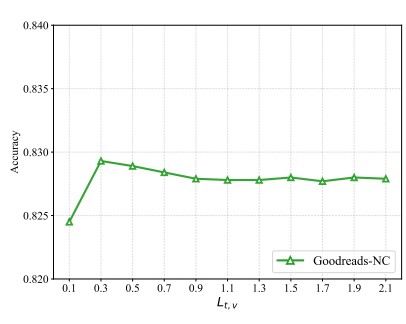

*Figure 12.* Goodreads-NC: $L_{tv}$ hyperparameter analysis.

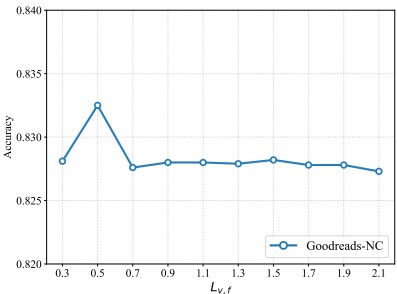

*Figure 13.* Goodreads-NC: $L_{vf}$ hyperparameter analysis.

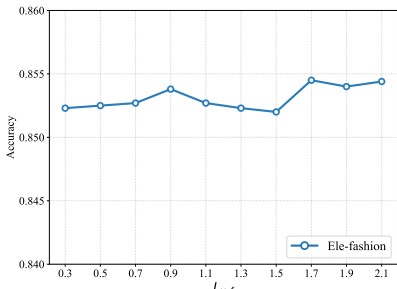

*Figure 14.* Ele-fashion: $L_{vf}$ hyperparameter analysis.

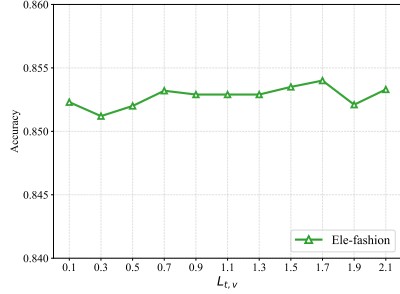

*Figure 15.* Ele-fashion: $L_{tv}$ hyperparameter analysis.

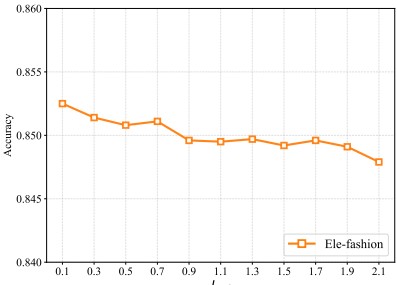

*Figure 16.* Ele-fashion: $L_{tf}$ hyperparameter analysis.

