# OpenReview forum: "A Graph Foundation Model with Cross-Modal  Alignment and Modality-Aware Expert Fusion for Multi-Modal Graphs"
_ICML.cc/2026/Conference — ICML 2026 regular_

### Official Review · Reviewer_Kudz · 2026-03-02

**Soundness:** 3
**Presentation:** 4
**Significance:** 4
**Originality:** 3
**Overall Recommendation:** 5
**Confidence:** 4

**Summary:**

This paper extends graph foundation models to multimodal graphs. The authors propose CAME, a multimodal graph foundation model for effective fusion and alignment of text and vision signals on graphs. CAME first fuses features using a dimension-wise gate and a modality-aware MoE, then aligns embeddings with cross-modal contrastive loss to learn a shared space. Experiments demonstrate that CAME outperforms existing methods across many datasets and tasks.

**Compliance With Llm Reviewing Policy:**

Affirmed.

**Key Questions For Authors:**

1. While the dim-gate is effective, I am curious how it behaves in practice. For instance, does the gate tend to assign high weights to one modality, or does it learn a more balanced mixture?

For other questions, see Weaknesses.

**Limitations:**

Yes

**Strengths And Weaknesses:**

Strengths
1. The paper makes a valuable contribution by extending GFM to multimodal data. The proposed model leverages multimodal graphs to not only enhance the performance of multimodal graphs but also on unimodal graphs. This further demonstrates that multimodal graphs are beneficial to the advancement of GFM.

2. The proposed method is novel and make sense. The modality-aware MoE and the tri-view cross-modal contrastive losses appear to adaptively integrate complementary signals across modalities.

3. The experiments are comprehensive and convincing, and the method performs strongly on both multimodal and unimodal scenarios.

Weaknesses
1. The method uses frozen CLIP as the entry point for text and image features. This shifts a part of the cross-modal alignment difficulty. It would be helpful to explain whether the improvement stems from the CLIP, or from the proposed fusion module.

2. The framework primarily targets two modalities. It is unclear if the method could generalize to three or more modalities, as the proposed method is multi-modal GFM rather than two-modal GFM.

3. Although ablations demonstrate that the dim-gate does works, it is still unclear what fusion behavior it learns in practice.

---

> ### Author Rebuttal · Authors · 2026-03-30
>
> We thank the reviewer for the careful reading of our paper. We provide our responses as follows.
>
> # For W1:
> Using frozen CLIP as the entry point partially alleviates the raw cross-modal gap. Our improvement does not mainly come from CLIP itself, but from the proposed structure-aware cross-modal alignment and fusion mechanism built on top of CLIP features.
>
> Although CLIP provides a shared embedding space, it does not eliminate the modality discrepancy in multimodal graphs. Even after CLIP encoding, text and image features still exhibit clear differences in both scale and distribution on the same dataset. This is exactly why CAME does not directly average or concatenate CLIP features. Instead, we first apply modality-specific graph encoders to preserve each modality’s structural patterns, and then perform dimension-wise gated fusion, modality-aware MoE fusion, and tri-view contrastive alignment among text, image, and fused views. These components are designed to address the modality discrepancy.
>
> We further analyze the impact of different modalities by evaluating CAME on multimodal graphs when only a single modality is effectively used as input. These results suggest that our method is not overly sensitive to whether one specific modality is stronger. In our additional single-modality analysis on multimodal graphs, CAME still achieves relatively stable performance when only one modality is used as input. This indicates that CAME does not merely rely on multimodal features.
>
> **Table 1. Impact of different input modalities on downstream tasks.**
> | Method | Ele-fashion | Amazon-cloth | Amazon-sports |
> |-|-|-|-|
> | CAME | **85.13 ± 0.03** | **37.68 ± 0.19** | **28.88 ± 0.12** |
> | Text-only | 84.90 ± 0.12 | 35.74 ± 0.08 | 27.97 ± 0.15 |
> | Image-only | 84.12 ± 0.10 | 32.58 ± 0.17 | 26.31 ± 0.16 |
> | CLIP-text | 83.53 ± 0.02 | 13.77 ± 0.09 | 25.13 ± 0.06 |
> | CLIP-image | 80.26 ± 0.05 | 13.09 ± 0.07 | 16.33 ± 0.02 |
>
>
> # For W2:
> We agree that the current experiments focus on two modalities, mainly because existing multimodal graph benchmarks only provide text and image attributes. However, CAME is not inherently restricted to two modalities. Its design is modular: each new modality can be incorporated by adding a modality-specific encoder and a corresponding expert group, while extending the alignment objective accordingly. Since these branches are organized in parallel, such an extension is computationally feasible. We will clarify in the revision that the current two-modality setting is due to dataset availability rather than a limitation of the framework, and leave validation on three or more modalities as future work.
>
> # For W3 and Q1:
> We appreciate the reviewer’s question. In practice, the gate weights remain stable during training: text changes from 0.48 to 0.42, image from 0.02 to 0.09, and fusion from 0.50 to 0.49. The image weight gradually increases, although it remains smaller than the text and fusion weights. This suggests that image features serve as complementary information to enrich the textual and fused representations. Therefore, the gate preserves contributions from multiple views through adaptively fusing multimodal features.
>
> More importantly, this empirical behavior is aligned with the overall objective of CAME: the gate is intended to softly balance complementary information across modalities, while the subsequent modality-aware MoE and tri-view alignment further stabilize the fused representation and reduce unimodal dominance. This is also supported by the ablation results, where replacing the dimension-wise gate with simple average pooling leads to consistent performance drops, indicating that the learned gate is providing meaningful adaptive fusion.
>
> We hope these clarifications adequately address your concerns. We are grateful for your feedback.

---

> > ### Author Rebuttal · Reviewer_Kudz · 2026-04-01
> >
> > My concerns have been addressed, and I will maintain my score.

---

> > > ### Author Response · Authors · 2026-04-05
> > >
> > > Thank you very much for your time and thoughtful suggestions. We are glad that our responses have addressed your concerns, and we sincerely appreciate your careful review.

---

### Official Review · Reviewer_cPnU · 2026-03-06

**Soundness:** 3
**Presentation:** 3
**Significance:** 2
**Originality:** 2
**Overall Recommendation:** 4
**Confidence:** 3

**Summary:**

This paper aims to address the issue in GFMs that focus solely on single-modal graphs. The model first generates structure-aware embeddings for each modality, then fuses multimodal information through a dimension-wise gated mechanism and a modality-aware MoE module. It further adopts a tri-view cross-modal contrastive loss to align text, vision, and fused embeddings. Extensive experiments demonstrate that the proposed model outperforms state-of-the-art GFMs.

**Compliance With Llm Reviewing Policy:**

Affirmed.

**Final Justification:**

My concerns have been addressed, and I will maintain my positive score.

**Key Questions For Authors:**

Please refer to the weakness.

**Strengths And Weaknesses:**

Strengths:
1. The proposed multimodal fusion framework combines dimension-wise gated fusion and modality-aware MoE with two-level routing, enabling adaptive fusion of multimodal features.
2. The experimental design is comprehensive and rigorous, involving ten diverse datasets, multiple evaluation settings (self-supervised, few-shot, in/out-of-domain), and in-depth analyses (ablation, hyperparameter sensitivity, visualization).
3. The paper is well-written and easy to follow.

Weaknesses:
1. The complexity analysis of the model only provides theoretical time complexity derivation, without conducting experimental comparisons of pretraining & inference speed and memory consumption with baseline models, making it difficult to evaluate its practical deployment feasibility on large-scale multimodal graphs.
2. There is a lack of experimental results and analysis regarding the impact of different modalities.
3. In addition to Unigraph2, there are also MGMF baselines (e.g., PLANET), and result comparison is lacking.

---

> ### Author Rebuttal · Authors · 2026-03-30
>
> We are grateful for your support and constructive suggestions. We respond to your suggestions point by point as follows.
>
> # For W1:
> We summarize GPU memory, pre-training time, and inference time in Table 1. To keep the comparison consistent, we fix the hidden size and batch size to 1024 for all methods. For sampling, GCOPE uses random-walk sampling, UniGraph2 and CAME use PPR-based sampling, and the other baselines use neighbor sampling [10, 5].
>
> As shown in Table 1, CAME remains efficient in practice. This efficiency comes from both the model design and the training pipeline. On the model side, CAME uses modality-specific propagation and a sparse top-k MoE, both of which can be executed in parallel. Its computational complexity also remains relatively low, since only a small number of experts are activated for each node. On the training side, we precompute and store PPR-sampled subgraphs offline, and further adopt subset pre-training on ogbn-Products. As a result, CAME achieves shorter pre-training time while maintaining competitive GPU memory usage and inference cost. The larger parameter count mainly comes from maintaining separate expert groups for text, image, and fusion views. We also conduct node classification on the large-scale ogbn-Products dataset (2,449,029 nodes and 61,859,140 edges), which further suggests that CAME is practical for large-scale graphs.
>
> **Table 1. Comparison of GPU memory usage, average joint pre-training time per epoch on ogbn-Products (large-scale dataset), Goodreads-LP, and Amazon-Cloth, downstream inference time for node classification on Goodreads-NC and parameters.**
> |Method |GPU Memory (GB)|Pre-training Time (min)|Inference Time (min)|Parameters (M)|
> |-|-|-|-|-|
> |**CAME**|**14.67**|**2.13**|**6.9**|**53.6**|
> |SAMGPT|15.64|23.04|7.1|5.3|
> |GCOPE|5.14|4.10|3.5|2.7|
> |FUG|18.45|13.09|11.2|10.5|
> |UniGraph2|28.37|11.58|10.3|32.5|
>
>
> # For W2:
> We further analyze the impact of different modalities by evaluating CAME on multimodal graphs when only a single modality is used as input. We replace the missing modality with the available one, thereby simulating missing-modality input.
>
> We observe three main findings from Table 2. First, removing either text or image leads to a performance drop compared with both modalities, which confirms that our fusion design effectively leverages multimodal information. Second, even under text-only or image-only input, CAME still outperforms CLIP-text and CLIP-image. This indicates that modality-specific graph propagation is effective, and that the gains do not come merely from stronger raw features, but also from graph structure. Third, these results further highlight the importance of graph structure: even when only one modality is available, CAME can still leverage structure-aware message passing to learn more transferable and discriminative representations.
>
> In addition, the performance drop is generally smaller in the text-only setting than in the image-only setting, suggesting that text usually provides richer semantic information for downstream tasks, while image mainly serves as a complementary modality.
>
> **Table 2. Impact of different input modalities on downstream tasks.**
> | Method | Ele-fashion | Amazon-cloth | Amazon-sports |
> |-|-|-|-|
> | **CAME** | **85.13 ± 0.03** | **37.68 ± 0.19** | **28.88 ± 0.12** |
> | Text-only | 84.90 ± 0.12 | 35.74 ± 0.08 | 27.97 ± 0.15 |
> | Image-only | 84.12 ± 0.10 | 32.58 ± 0.17 | 26.31 ± 0.16 |
> | CLIP-text | 83.53 ± 0.02 | 13.77 ± 0.09 | 25.13 ± 0.06 |
> | CLIP-image | 80.26 ± 0.05 | 13.09 ± 0.07 | 16.33 ± 0.02 |
>
>
> # For W3:
> Thank you for pointing out this valuable related work. PLANET is a very recent and important work, released on arXiv on February 4, 2026, which is after the ICML submission deadline. It advances multimodal graph modeling by addressing insufficient modality interaction and suboptimal alignment through a divide-and-conquer framework at both the embedding and node levels.
> Since PLANET was released very recently, and its official code seems not to be publicly available at present, it is difficult to reproduce fairly within the short rebuttal period. For this reason, we are currently unable to provide a reliable comparison.
>
> Thank you for this helpful suggestion. We will cite and discuss PLANET explicitly as an important MGFM in the revised version, and we will try our best to include an experimental comparison in future versions.
>
> We sincerely thank you again for your insightful comments, which prompted us to reflect more deeply on our work. We hope these clarifications address your concerns. For any other questions, we are happy to discuss them and make further clarifications.

---

> > ### Author Rebuttal · Reviewer_cPnU · 2026-04-03
> >
> > My concerns have been addressed, and I will maintain my score.

---

> > > ### Author Response · Authors · 2026-04-05
> > >
> > > Many thanks for your time and insightful suggestions, which have greatly helped us improve our work. We are glad that our responses have helped address your concerns, and we would be very happy to provide further clarification if needed.

---

### Official Review · Reviewer_SkV4 · 2026-03-10

**Soundness:** 3
**Presentation:** 3
**Significance:** 3
**Originality:** 3
**Overall Recommendation:** 5
**Confidence:** 3

**Summary:**

This paper investigates multi-modal graph foundation models, pointing out that existing GFMs primarily target unimodal graphs, whereas real-world scenarios often involve nodes possessing multimodal attributes such as text and images. Furthermore, existing multi-modal GFMs like UniGraph2 average raw embeddings from different modalities before modeling, potentially causing information loss prior to model input and hindering effective cross-modal alignment and fusion. Considering that different scenarios often exhibit distinct modal patterns, the authors note that imposing a shared encoding mechanism across all modalities risks overlooking these variations and failing to recognize differentiated contributions. To address this, they propose CAME: first performing modality-specific graph encoding, then fusing representations through dimension-wise gated fusion and modality-aware MoE, while aligning text, image, and fused representations using contrastive loss. Experiments encompassed self-supervised pretraining, few-shot transfer learning, ablation studies, and loss-weight sensitivity analysis, reporting improvements over UniGraph2 and several GFM/SSL baselines.

**Compliance With Llm Reviewing Policy:**

Affirmed.

**Final Justification:**

The authors’ response has addressed my concerns, and I will maintain my score.

**Key Questions For Authors:**

1.Can you provide more equitable comparison metrics? This should include the number of parameters for each model, whether CLIP features were frozen, and why “CLIP+concat” is sufficient to represent baselines that originally did not support multimodal capabilities?
2.Regarding the accuracy comparison with UniGraph2. The paper describes UniGraph2 as “merely averaging raw embeddings before modeling,” but in reality, UniGraph2 performs averaging after modality-specific encoding and before inputting to MoE. Could the authors further clarify the specific technical differences between CAME and UniGraph2?
3.While complexity analysis is provided in the appendix, given that the title positions this method as a Foundation Model, could the authors further elaborate on the actual differences between CAME and baseline models in terms of parameter scale, training time, and memory consumption? Additionally, how does the method's training scalability perform on larger-scale graph data, such as its training efficiency and resource requirements when node or edge scales increase further? Providing more concrete discussions on these aspects may help readers gain a more comprehensive understanding of the method's applicability as a Foundation Model.

**Limitations:**

yes

**Strengths And Weaknesses:**

Strengths
1.The problem is significant and timely. The paper focuses on extending GFMs to multi-modal graphs, an inherently meaningful direction; the authors also clearly pinpoint the core challenges as “co-encoding structural and multi-modal features”and“cross-modal alignment/fusion.”
2.The method design aligns well with the problem formulation. The paper's three core components—modality-specific GNNs, dimension-level gates, modality-aware MoE, and three-view alignment—all intuitively address the authors' identified issue of “information loss due to coarse-grained fusion / modality dominance.”The methodological chain is relatively complete, with clear functional divisions among the modules.
3.The experiments are extensive, yielding robust results. In the main results table, CAME outperforms UniGraph2 and other baselines in both self-supervised representation learning and few-shot transfer. Ablation studies demonstrate that m-GNN, dim-gate, MoE, and tri-view losses all contribute to the performance. Empirical evidence provides substantial support for the paper's claims.

Weaknesses
1.The innovation holds practical significance but remains relatively limited in scope. The authors introduce modality-specific propagation + dimension-wise gating + modality-aware MoE + tri-view alignment. This direction of improvement is reasonable and may indeed yield better performance. However, from an innovation perspective, it appears more as an integration and refinement of existing multimodal image modeling components rather than a wholly novel methodological paradigm.
2.The introduction does not sufficiently cover UniGraph2, which, as the most relevant prior work, should be more accurately positioned within the introduction. Additionally, the abstract uses “modality-aware Expert fusion,” while the main text occasionally refers to “modality-aware MoE” and at other times to “multimodal MoE.” We recommend standardizing the terminology.
3.Experimental Considerations. The authors note that, except for UniGraph2, other models do not natively support multi-modal input. Therefore, they uniformly concatenate inputs after encoding them with CLIP. For text-only datasets, CLIP text embeddings are used to represent the missing image modality. While this approach is reasonable, it raises the question of whether it compromises the original models' performance, potentially leading to comparisons that are not entirely native or conducted under equivalent capability boundaries.

---

> ### Author Rebuttal · Authors · 2026-03-30
>
> Thank you for your thoughtful and positive feedback on our work. We respond to your concerns as follows.
>
> # For W1:
> Yes, indeed, part of the used techniques have been explored in other domains, while our contribution lies in addressing the challenge of coupling multimodal data fusion and alignment with graph structure propagation in multimodal graphs. This problem becomes more challenging with the introduction of graph structure and remains underexplored in prior work. Through our design, these existing techniques are effectively integrated to mitigate this challenge. In addition, as one of the early attempts in MGFMs, we hope this work can serve as a useful foundation for future advances in MGFMs. Thanks for your valuable suggestions, we will explore new methodological paradigms in future work to further enhance MGFMs.
>
> # For W2 & Q2:
> Thanks. We apologize for the unclear description of UniGraph2. UniGraph2 is an important early work and one of the pioneering attempts in MGFMs. After modality-specific encoding, UniGraph2 averages the CLIP-encoded features before structure propagation for modality fusion, which makes it difficult to adaptively preserve modality-specific structural information. In contrast, CAME preserves separate modality branches during graph propagation, and then applies dimension-wise gated fusion, modality-aware MoE, and tri-view contrastive alignment. This enables finer-grained fusion and alignment across dimensions, nodes, modalities, and domains. We will revise the paper to describe UniGraph2 more precisely and clarify this distinction more explicitly.
>
> Besides, thank you for pointing out the terminology issue. We will use “modality-aware expert fusion” consistently throughout the paper.
>
> # For W3 and Q1:
> There might be some misunderstanding. We further clarify the setup for models that do not natively support multimodal input (GCOPE, SAMGPT, and FUG). **On text-only graphs**, we directly use the CLIP-encoded text feature as the input, without concatenation. This is because these models are originally designed for text inputs or handcraft features, rather than multimodal graphs. Such treatment helps them achieve their best performance, thereby supporting a fair comparison. **On multimodal graphs**, we use the concatenation of CLIP text and image embeddings for these baselines. We adopt this strategy because text and image features exhibit noticeable differences in distribution  and it is a standard protocol in multimodal graph benchmarks. Direct addition or averaging may blur modality-specific semantics, whereas concatenation better preserves information from both modalities.
>
> The CLIP encoder is frozen for all settings. This ensures that the comparison is not affected by differences in CLIP fine-tuning and also avoids the substantial additional time cost of jointly pretraining the graph model and CLIP. This choice is also consistent with UniGraph2.
>
> In addition, following your suggestion, we report the parameters of the graph foundation models: GCOPE: 2.7M, SAMGPT: 5.3M, FUG: 10.5M, UniGraph2: 32.5M, CAME: 53.6M. Although CAME has a relatively larger parameter count, this mainly comes from assigning a dedicated expert group to each modality. As noted in our response to Q3, this does not lead to a proportional increase in practical cost.
>
> # For Q3:
> We report GPU memory usage, pre-training time, inference time and parameter scale. For a fair comparison, we set the hidden dimension and batch size to 1024. GCOPE uses its original random-walk sampling, UniGraph2 and CAME use PPR sampling, and the other baselines use neighbor sampling [10, 5].
> As shown in Table 1, CAME remains efficient, with shorter pre-training time and competitive GPU memory usage and downstream inference time. We attribute this efficiency to its low computational complexity and several design choices, including offline precomputed PPR subgraphs, subset pre-training on ogbn-Products, and sparse top-k MoE routing. As for parameters, CAME has a relatively larger parameter count, mainly because it assigns a dedicated expert group to each modality. Regarding scalability on large graphs, we have conducted node classification experiments on ogbn-Products, which contains 2,449,029 nodes and 61,859,140 edges, demonstrating the capability of CAME to handle large-scale graphs.
>
> **Table 1. Comparison of GPU memory usage, average joint pre-training time per epoch on ogbn-Products (**large-scale dataset**), Goodreads-LP, and Amazon-Cloth, downstream inference time for node classification on Goodreads-NC and parameters.**
> |Method | GPU Memory (GB) | Pre-training Time (min) |Inference Time (min)|Parameters (M)|
> |-|-|-|-|-|
> |**CAME**|**14.67**|**2.13**|**6.9**|**53.6**|
> |SAMGPT|15.64|23.04|7.1|5.3|
> |GCOPE|5.14|4.10|3.5|2.7|
> |FUG|18.45|13.09|11.2|10.5|
> |UniGraph2|28.37|11.58|10.3|32.5|
>
> Thanks again for your constructive comments. Please feel free to contact us if you have any other concerns.

---

> > ### Author Rebuttal · Reviewer_SkV4 · 2026-04-01
> >
> > The authors’ response has addressed my concerns, and I will maintain my score.

---

> > > ### Author Response · Authors · 2026-04-05
> > >
> > > We are pleased that your concerns have been addressed. We also greatly appreciate your positive view of our work and we will continue to incorporate your thoughtful feedback going forward.

---

### Official Review · Reviewer_6UDE · 2026-03-12

**Soundness:** 3
**Presentation:** 2
**Significance:** 3
**Originality:** 2
**Overall Recommendation:** 4
**Confidence:** 3

**Summary:**

In this paper, the authors propose CAME, a framework designed to jointly model graph structures and multimodal node features. The method introduces modality-specific graph encoders, a dimension-wise gating mechanism for multimodal fusion, a modality-aware Mixture-of-Experts (MoE) module, and uses a cross-modal contrastive objective to align representations across modalities. Experiments on multiple multimodal graph datasets demonstrate improved performance compared to several graph foundation model baselines.

**Compliance With Llm Reviewing Policy:**

Affirmed.

**Final Justification:**

I main my original score.

**Key Questions For Authors:**

1. The model includes modality-specific encoders, dimension-wise gating, and MoE routing. How about the computational overhead compared to standard GFMs?

2. Real-world multimodal graphs often contain missing or noisy modalities. How robust is CAME when one modality is missing or corrupted for some nodes?

**Limitations:**

No limitation discussion

**Strengths And Weaknesses:**

Strengths:

1. Extending GFMs to handle multimodal graphs is a meaningful direction, as many real-world graphs naturally contain heterogeneous modalities such as text and images.

2. The experiments on multiple multimodal graph datasets across several settings (in-distribution, in-domain, and out-of-domain) demonstrate improved performance over several GFM baselines.

3. The t-SNE visualizations provide useful qualitative evidence that the proposed method can learn more transferable and separable representations.

Weaknesses:

1. Although the paper focuses on multimodal graph foundation models, many of the techniques used in the framework including Mixture-of-Experts (MoE) routing, modality-specific encoders, gating mechanisms, and cross-modal contrastive learnin have been extensively studied in prior work. As a result, the level of novelty mainly lies in integrating these components into the graph foundation model setting.

2. The model includes several potentially expensive components. However, the paper does not provide a detailed analysis of training cost, memory usage, or inference efficiency, which is important for evaluating scalability to large graphs.

3. Although the paper introduces a modality-aware MoE module, the empirical analysis of expert routing remains limited. Several aspects of the module are not fully characterized, including the specialization of different experts to distinct modality patterns, the stability of routing behavior across datasets, and the frequency of expert activation during training and inference.

---

> ### Author Rebuttal · Authors · 2026-03-30
>
> We sincerely thank you for your support of our work and your valuable comments. Our responses are provided below.
>
> # For W1:
> We agree that this is a reasonable concern. Our main contribution lies in making multimodal fusion, alignment, and graph propagation work together in a unified method. The novelty of our work is performing modality-specific graph propagation for different modalities, while enabling comprehensive fusion at the dimension, node, and cross-graph levels. As one of the early attempts in this direction, we hope this work can contribute to future progress in MGFMs.
>
> # For W2 and Q1:
> Table 1 reports GPU memory usage, pre-training time, inference time and parameters. For a fair comparison, we unified the hidden dim. It has shorter pre-training time, while maintaining competitive GPU memory usage and inference efficiency. This is enabled by several acceleration designs, including offline precomputed PPR subgraphs, subset pre-training on ogbn-Products, and sparse top-k MoE routing. The extra parameters of CAME mainly come from modality-specific expert groups. To verify scalability, we conducted node classification on the large-scale ogbn-Products graph (2,449,029 nodes, 61,859,140 edges). This proves that CAME can be applied to large-scale graphs.
>
> **Table 1. Comparison of GPU memory, average joint pre-training time per epoch on ogbn-Products (large-scale dataset), Goodreads-LP, and Amazon-cloth, downstream inference time on Goodreads-NC and parameters.**
> |Method|GPU Memory (GB)|Pre-training Time (min)|Inference Time (min)|Parameters (M)|
> |-|-|-|-|-|
> |**CAME**|**14.67**|**2.13**|**6.9**|**53.6**|
> |SAMGPT|15.64|23.04|7.1|5.3|
> |GCOPE|5.14|4.10|3.5|2.7|
> |FUG|18.45|13.09|11.2|10.5|
> |UniGraph2|28.37|11.58|10.3|32.5|
>
> # For W3:
> Thanks. We clarify and analyze the MoE used in CAME. We assign an expert group to each modality and design a gating strategy for cross-modal fusion. The gate weights remain stable during training: text changes from 0.48 to 0.42, image from 0.02 to 0.09, and fusion from 0.50 to 0.49. Although the image weight is smaller, its gradual increase suggests that image features provide complementary information.
>
> We further analyze expert activation frequency during the early stage of pre-training and inference, as shown in Tables 2 and 3. Different datasets already exhibit distinct expert activation patterns across text, image, and fusion groups during training. During inference, the model further selects the most suitable experts for each dataset, and these selections are not always identical to the early-stage activation patterns. This indicates that CAME can train experts adequately during pre-training and adaptively route different inputs to appropriate experts during inference, enabling domain-aware and modality-aware specialization.
>
> **Table 2. Expert Activation Frequency During Pre-training Across Datasets and Modalities.**
>
> |Expert Type|Dataset|E1|E2|E3|E4|E5|E6|
> |-|-|-|-|-|-|-|-|
> |Text|ogbn-Products|0.5|1.2|35.0|31.3|6.8|25.2|
> |Text|Goodreads-LP|0.6|0.2|33.8|36.1|6.3|23.1|
> |Text|Amazon-cloth|0.3|0.1|33.1|27.0|6.3|33.1|
> |Image|ogbn-Products|11.1|43.4|1.2|1.1|31.0|12.2|
> |Image|Goodreads-LP|18.6|26.4|2.0|0.7|14.7|37.8|
> |Image|Amazon-cloth|26.3|15.3|1.0|0.2|13.6|43.6|
> |Fusion|ogbn-Products|38.7|10.7|2.3|0.7|15.9|31.7|
> |Fusion|Goodreads-LP|37.4|8.6|6.6|2.3|13.3|31.7|
> |Fusion|Amazon-cloth|36.3|11.0|3.9|3.1|14.1|31.6|
>
> **Table 3. Selected Experts at Inference Across Datasets and Modalities.**
> |Expert Type|Dataset|Selected|
> |-|-|-|
> |Text|Amazon-cloth|E3,E4|
> |Text|Arxiv|E4,E6|
> |Text|WN18RR|E1,E2|
> |Image|Amazon-cloth|E1,E6|
> |Image|Arxiv|E3,E4|
> |Image|WN18RR|E2,E6|
> |Fusion|Amazon-cloth|E1,E6|
> |Fusion|Arxiv|E2,E3|
> |Fusion|WN18RR|E2,E6|
>
> # For Q2:
> CAME is also robust to missing modalities, as shown in Table 1 of our paper on text-only datasets such as Arxiv and WikiCS. For such modality-missing datasets, we replace the missing image modality with the available one. CAME still consistently improves downstream performance.
>
> We further evaluate the robustness of CAME under modality corruption by injecting Gaussian noise with different ratios into Ele-fashion. From Table 4, CAME remains robust under modality corruption. Its performance degrades gradually as the noise ratio increases, and the largest drop occurs when both modalities are corrupted. At the same time, text-only noise causes a larger drop than image-only noise, suggesting that text is the more informative modality on this dataset, while image mainly plays a complementary role.
>
> **Table 4. Modality Corruption Results on Ele-fashion.**
> |Modality|0.1|0.2|0.3|0.4|0.5|
> |-|-|-|-|-|-|
> |Text-only noise|84.43 ± 0.05|84.00 ± 0.06|83.61 ± 0.23|83.17 ± 0.10|82.73 ± 0.11|
> |Image-only noise|84.75 ± 0.10|84.59 ± 0.12|84.48 ± 0.13|84.34 ± 0.10|84.08 ± 0.10|
> |Both-modalities noise|84.30 ± 0.06|83.48 ± 0.08|82.88 ± 0.11|82.34 ± 0.08|81.38 ± 0.22|
>
> Thanks again for your constructive comments. We hope these clarifications address your concerns.

---

> > ### Author Rebuttal · Reviewer_6UDE · 2026-04-04
> >
> > My main concerns have been addressed, I will maintain my score.

---

> > > ### Author Response · Authors · 2026-04-05
> > >
> > > We are deeply grateful for the reviewer’s time and constructive feedback. We are pleased that our responses have helped address your concerns. If there are still any questions that you would like us to clarify further, we would be very happy to do so.

---

### Decision · Program_Chairs · 2026-04-30

**Decision:**

Accept (regular)

**Comment:**

The paper proposes a multimodal graph foundation model called CAME. The framework encodes each modality with graph embeddings, fuses them with a dimension-wise gating mechanism and modality-aware MoE routing, and finally aligns them using a contrastive loss.

The paper was well-received by the reviewers, which mainly praised the timeliness of this work, the architectural coherence and plausibility, and the broad empirical evaluation. One important point of strength is that the model seems to work well also in unimodal scenarios.

Concerns revolved mostly around whether the method is practical and scalable (e.g. missing information on training/MoE costs), some confusing positioning (e.g. was not really clear how CAME distinguishes from UniGraph2), and a missing comparison with PLANET. These concerns can be considered solved after the rebuttal: scalability and positioning were cleared out, while the missing comparison is inapplicable because PLANET was published after the paper was submitted.

A third concern raised by reviewers is about novelty: the paper reads like a well-engineered ensemble of known techniques than a methodological breakthrough. The authors argued that bringing together CAME components for multimodal graph foundation models and doing modality-specific fusion and alignment can be considered novel. While their answer is reasonable, it only partially clears up the novelty concern.

In my opinion, while it does not fully check the mark of groundbreaking technical novelty, this paper provides a solid contribution to an emerging and important research topic. Therefore, I recommend acceptance.